 RESEARCH COMMUNICATION

# Disparate expression specificities coded by a shared Hox-C enhancer

**Steve W Miller\*, James W Posakony**

Division of Biological Sciences, Section of Cell & Developmental Biology, University of California San Diego, La Jolla, United States

**Abstract** Can a single regulatory sequence be shared by two genes undergoing functional divergence? Here we describe a single promiscuous enhancer within the *Drosophila* Antennapedia Complex, EO053, that directs aspects of the expression of two adjacent genes, *pb* (a *Hox2* ortholog) and *zen2* (a divergent *Hox3* paralog), with disparate spatial and temporal expression patterns. We were unable to separate the *pb*-like and *zen2*-like specificities within EO053, and we identify sequences affecting both expression patterns. Importantly, genomic deletion experiments demonstrate that EO053 cooperates with additional *pb*- and *zen2*-specific enhancers to regulate the mRNA expression of both genes. We examine sequence conservation of EO053 within the Schizophora, and show that patterns of synteny between the *Hox2* and *Hox3* orthologs in Arthropods are consistent with a shared regulatory relationship extending prior to the *Hox3/zen* divergence. Thus, EO053 represents an example of two genes having evolved disparate outputs while utilizing this shared regulatory region.

**Editorial note**: This article has been through an editorial process in which the authors decide how to respond to the issues raised during peer review. The Reviewing Editor's assessment is that all the issues have been addressed (see decision letter).

**\*For correspondence:**
swmiller@ucsd.edu

**Competing interests:** The authors declare that no competing interests exist.

## Introduction

Changes in the expression specificity of genes involved in the development of multicellular organisms are implicated in modifications of form and function over evolution (*Stern and Orgogozo, 2008*; *Wray, 2007*; *Rebeiz and Tsiantis, 2017*; *Rubinstein and de Souza, 2013*). To produce these distinct expression patterns, the promoters of many developmental genes are activated in specific spatiotemporal domains by one or more distal *cis*-regulatory sequences (*Long et al., 2016*; *Levine, 2010*). Over the last three decades, two contrasting modes of promoter regulation by such sequences have emerged. Commonly, a gene specifically expressed in multiple diverse developmental contexts has distinct *cis*-regulatory sequences known as enhancers, each of which directs expression in a specific, limited subset of the overall context (*Kuzin et al., 2012*; *Frost et al., 2018*; *Simonet et al., 1991*; *MacNeill et al., 2000*; *Harding et al., 1989*). In a second mode, multiple neighboring genes with overlapping expression domains can be controlled by a shared distal *cis*-regulatory region, referred to as a locus control region (LCR), that directs expression of the target genes in a common spatial and temporal pattern during development (*Ahn et al., 2014*; *Choi and Engel, 1988*; *Deschamps, 2007*; *Foley et al., 1994*; *Lehoczky et al., 2004*; *Sharpe et al., 1998*; *Spitz et al., 2003*; *Tsai et al., 2016*; *Jones et al., 1995*; *Tsujimura et al., 2007*; *Mohrs et al., 2001*). These two modes of activation are not mutually exclusive, and genes regulated by LCRs can also have their own independent enhancers (*Deschamps, 2007*; *Jones et al., 1995*).

Most experimental models for changes in patterns of gene expression have come from studies of specific enhancers. The developmental context of an enhancer's action is typically determined by the sequence-directed recruitment of specific DNA-binding transcription factors (*Levine, 2010*). The specificity of an enhancer can be modified in evolution by DNA mutations affecting the complement

of transcription factors recruited to the module (*Glassford and Rebeiz, 2013*; *Rebeiz and Tsiantis, 2017*; *Stern and Frankel, 2013*). Thus, enhancers can acquire additional specificities that change the expression pattern of their target genes as long as the change is either not detrimental or accompanied by additional stabilizing mutations. Such models for enhancer evolution are often proposed in the context of 'shadow enhancers', in which two enhancers regulating the same gene have overlapping and/or synergistic activity (*Barolo, 2012*; *Perry et al., 2011*; *Miller et al., 2014*; *Cannavò et al., 2016*; *Stern and Frankel, 2013*). In this case, the partial redundancy between the two enhancers could buffer the effects of mutation and divergence of regulatory sequence (*Payne and Wagner, 2015*). When an enhancer acquires multiple specificities, evolution can potentially lead to 1) loss of the newly acquired specificity (*Jeong et al., 2008*; *Jeong et al., 2006*), 2) loss of the original specificity, 3) complete loss of enhancer function, or 4) maintenance of the complex pattern. The latter two outcomes may be dependent upon the degree of use of the same transcription factors for both specificities, as loss of binding sites for shared factors would affect both expression patterns (*Rebeiz et al., 2011*).

In this work, we investigate an unusual case of complex expression through analysis of a 1.4-kb enhancer, referred to as EO053. We identified this region through the modENCODE effort, based upon detection of CBP binding only during embryonic stages ('Embryo Only 053') by chromatin immunoprecipitation (*Nègre et al., 2011*). We show that EO053 encodes complex spatiotemporal activity correlating with the evolutionary divergence in the expression and function of the two neighboring developmental genes under its regulatory influence. We present a distinctive mode of activation by EO053, in which each target gene utilizes EO053 for distinct spatiotemporal outputs.

EO053 is located within an intron of the *proboscipedia* (*pb*) gene in *Drosophila melanogaster*, which encodes a homeodomain-containing transcription factor involved in patterning along the anteroposterior axis. *pb* is found within a complex of related homeobox (Hox) genes, a pattern common in metazoans (*Lemons and McGinnis, 2006*). This collection of genes in *D. melanogaster*, referred to as the Antennapedia Complex (Antp-C), represents half of an ancestral Arthropod Hox gene complex that bifurcated within the Schizophora clade of flies into the Antp-C and Bithorax Complex (Bx-C), located 10 megabases away (*Negre and Ruiz, 2007*). Adjacent to *pb*, which is the *Hox2* ortholog, are three genes derived from the ancestral Arthropod *Hox3* gene: *zerknüllt* (*zen*), its duplicate *zen2*, and *bicoid* (*bcd*). At an early stage of insect evolution, the *Hox3* ortholog (*zen*) diverged in both expression and function away from anteroposterior patterning to specifying extra-embryonic tissue at earlier stages in embryonic development (*Hughes et al., 2004*). More recently within Schizophoran flies, tandem duplications of *zen* produced *zen2* and *bcd* (*Stauber et al., 1999*; *Stauber et al., 2002*; *Negre et al., 2005*), the latter of which diverged further into a role as a morphogen specifying the anterior pole of the embryo (*Driever and Nüsslein-Volhard, 1988*; *Struhl et al., 1989*). The insect radiation that followed the *Hox3*/*zen* divergence dates to the Devonian period (*Misof et al., 2014*), implying that the regulatory changes in *zen* are roughly 400 million years old. Intriguingly, EO053 encodes a union of expression patterns resembling both *pb* and its immediate upstream neighbor, *zen2*, suggesting that this enhancer may regulate the expression of both genes even though they are activated in unrelated, non-overlapping tissues and developmental stages. Here we show that deletion of EO053 via CRISPR/Cas9 affects mRNA accumulation from both *pb* and *zen2*, indicating that indeed this enhancer is shared between these two genes. We also find that the sequences responsible for the *pb*-like and *zen2*-like expression patterns within EO053 are highly overlapping, and we identify nucleotide segments that contribute to both specificities. One such nucleotide block contains a conserved sequence existing prior to the Schizophoran *zen*-*zen2* duplication, and we find that variants of this motif exhibit patterns of conservation within many of the major insect clades following the *Hox3*/*zen* divergence. Finally, we show that the pattern of synteny between *zen* and *pb* within the insects, and the lack of separation of these genes by translocation, is consistent with an ancient regulatory relationship between them, even in the face of disparately evolving specificities.

## Results

### The EO053 enhancer specificities overlap the expression patterns of both *pb* and *zen2*

The location of EO053 within the large intron of *pb* suggested that *pb* itself may be the target of this enhancer (*Figure 1A*). Indeed, at embryonic stages when Pb protein is detected in maxillary and labial segments (*Pultz et al., 1988*) we find that EO053 drives *GAL4* expression in these territories as well (*Figure 1E-G*), although not in a pattern as expansive as that driven by the previously-studied *pb* regulatory region 2.1 (*Figure 1A*; *Kapoun and Kaufman, 1995a*). Interestingly, in blastoderm stages EO053 also drives *GAL4* expression dorsally along most of the length of the embryo (similar to the pattern seen in *Figure 1B*). Following gastrulation, *GAL4* is detected in the amnioserosa (*Figure 1C,D*), a specificity derived from the earlier dorsal cell population (*Hartenstein, 1995*). This pattern is not representative of *pb*, but rather mimics the expression of *zen2* (*Pultz et al., 1988*), the gene immediately upstream of *pb* (*Figure 1A*). This additional expression could simply represent a

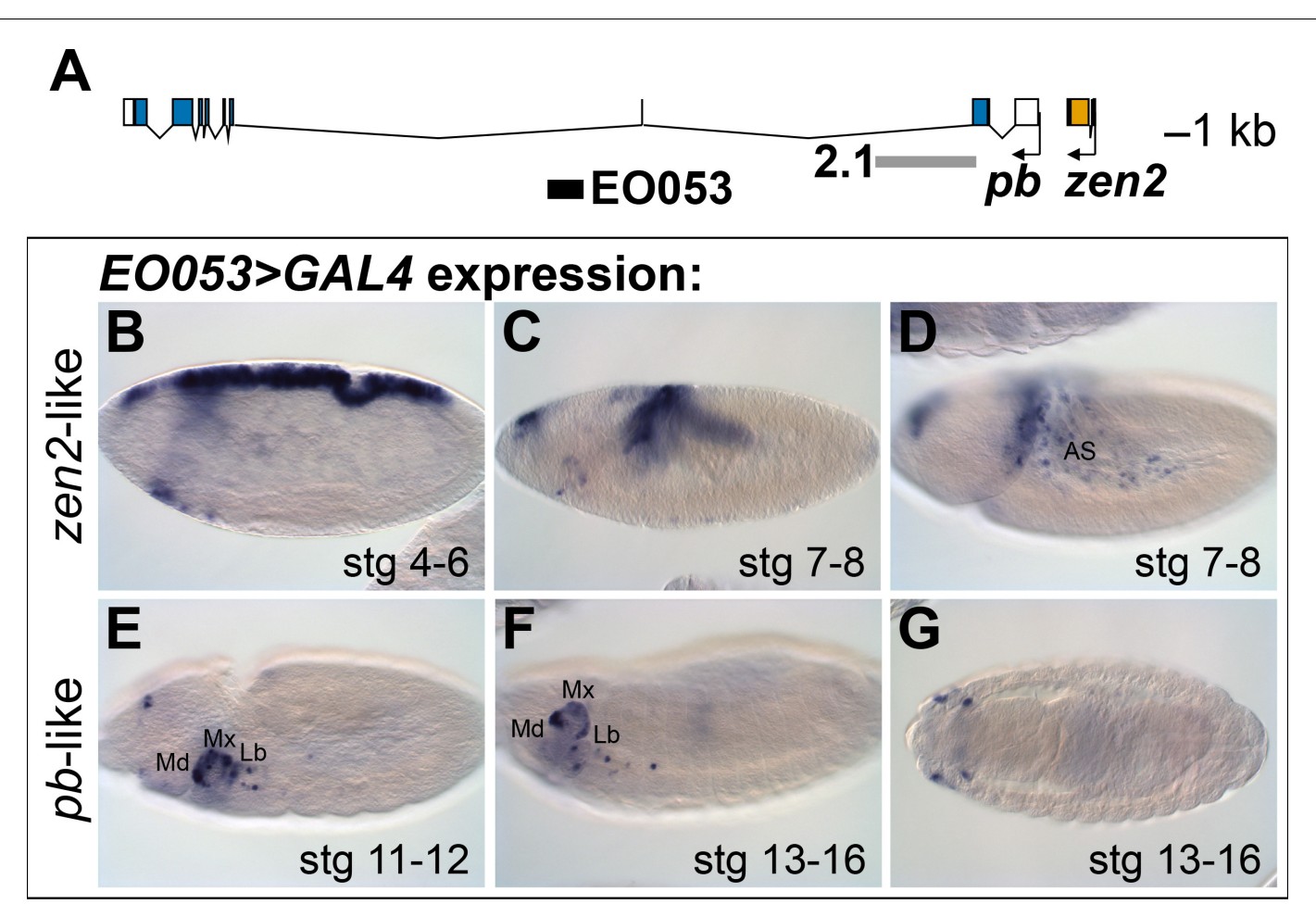

**Figure 1.** EO053 exhibits both *zen2*-like and *pb*-like expression patterns. (A) Diagram of the *pb* (blue) and *zen2* (yellow) genes and the locations of EO053 (black bar) and the 2.1 *pb* regulatory region (grey bar) (*Kapoun and Kaufman, 1995a*). Scale is shown at upper right. (B-G) Expression of *GAL4* mRNA by in situ hybridization in *EO053>GAL4* embryos exhibits a pattern reminiscent of *zen2* (*Rushlow et al., 1987*) in early embryonic stages (B-D; see also *Figure 5* and http://insitu.fruitfly.org/cgi-bin/ex/report.pl?ftype=1&ftext=FBgn0004054) and overlaps expression of *pb* (*Pultz et al., 1988*) in later stages (E-G; see also *Figure 5* and http://insitu.fruitfly.org/cgi-bin/ex/report.pl?ftype=1&ftext=FBgn0051481). AS: amnioserosa. Md: mandibular segment. Mx: maxillary segment. Lb: labial segment. See also *Figure 1—figure supplement 1*.

The online version of this article includes the following figure supplement(s) for figure 1:

**Figure supplement 1.** Summary of Reporter Fragments.

coincidental ectopic artifact of the precise genomic segment chosen for cloning the enhancer. However, the orthologous region from *Drosophila virilis* also encodes both expression specificities, reducing the likelihood of this being a chance occurrence (*Figure 1—Figure Supplement 1A, C – C''*).

While numerous regulatory regions have been shown to serve more than one promoter (*Choi and Engel, 1988*; *Deschamps, 2007*; *Foley et al., 1994*; *Lehoczky et al., 2004*; *Sharpe et al., 1998*; *Spitz et al., 2003*; *Tsai et al., 2016*; *Jones et al., 1995*; *Tsujimura et al., 2007*; *Mohrs et al., 2001*), these genes typically have expression specificities in common. EO053, then, may serve as an example of a regulatory region that serves more than one promoter but with each gene utilizing the region to generate a different specificity. We thus sought to determine how linked are these specificities and whether each gene indeed requires the EO053 region for expression.

## The *pb*-like expression specificity derives from the central region of EO053

We first began analyzing EO053 under a simple model for encoding multiple specificities: Each expression pattern is dependent upon a separate subregion of the 1.4-kb EO053 sequence. We created a set of reporter constructs containing overlapping truncated portions of EO053 (trunc1, trunc2, trunc3, trunc1-2, trunc2-3; *Figure 2*). The central region, trunc2, drives both *pb*-like and *zen2*-like expression but not as robustly as the full EO053 construct (*Figure 2E,F*). This region also drives ectopic expression in the ventral embryo at stage 10 (*Figure 2F*) and ectopic dorsal expression (amnioserosa or dorsal vessel) in late-stage embryos (*Figure 2G*). The right-most region, trunc3, also drives *pb*-like expression, though very weakly (*Figure 2I,J*). A construct that encompasses both the trunc2 and trunc3 regions drives reporter expression in a robust *pb*-like pattern that also lacks the ectopic activities seen with trunc2 alone (*Figure 2O,P*), suggesting that trunc3 contains elements that repress the late dorsal expression. This model is further supported by the trunc1-2 construct (removing the right-most portion of EO053) that also drives ectopic late dorsal expression (*Figure 2M*).

## The *zen2*-like and *pb*-like expression specificities are not easily separable

While trunc2 retains some capacity to drive *zen2*-like expression (*Figure 2E*), it was only weakly detectable in a few early-gastrulation embryos (embryo in *Figure 2E* is rare; most stage 5 embryos lack *GAL4* expression). A construct including trunc2 and the left-most portion of EO053, trunc1-2, restores *zen2*-like expression (*Figure 2K*), yet the left-most portion alone, trunc1, fails to drive reporter expression at any stage (*Figure 2B–D*). Since these initial truncation constructs failed to reveal a region in EO053 responsible for the *zen2*-like expression, we designed a set of 10 smaller overlapping reporter constructs to locate the *zen2*-like activity (*Figure 3A*). None of these smaller fragments drive reporter expression in a *zen2*-like pattern (*Figure 3—figure supplement 1*), while two fragments, truncF and truncG, drive reporter expression in a *pb*-like pattern (*Figure 3C,D,F,G*), consistent with their overlap with trunc2 (*Figure 3A*). Furthermore, we again observed late expression in the dorsal embryo driven by truncG (*Figure 3G*), refining the location of this ectopic activity seen neither with full-length *EO053>GAL4* or endogenous *pb* or *zen2* mRNA. These fragments allowed us to further define the domain sufficient to produce the *pb*-like pattern, and suggested that if the *pb*-like and *zen2*-like specificities were separable, the latter pattern would be localized to the left-most region of EO053 outside of truncF and truncG. Importantly, truncF-J, which lacks this left-most region, fails to drive *zen2*-like *GAL4* expression (*Figure 3K*). However, truncA-D, a construct containing only this region, fails to drive strong *zen2*-like reporter expression (*Figure 3H*). This suggests that while the A-D region is necessary (but not sufficient) for the *zen2*-like pattern, the FG region likely also contains elements required for this specificity, in addition to being sufficient to drive *pb*-like expression. Consistent with this interpretation, a construct that deletes this region, truncΔFG, lost both *pb*-like and *zen2*-like expression patterns (*Figure 3N–P*).

Because the FG region is necessary for both the *pb*-like and *zen2*-like patterns, we sought an alternative approach to determine if the two patterns are indeed separable. In a series of eight constructs, we created successive 47-nt non-complementary transversion mutations along the length of the FG region to identify sequences required for either the *pb*-like or *zen2*-like patterns (*Figure 4*).

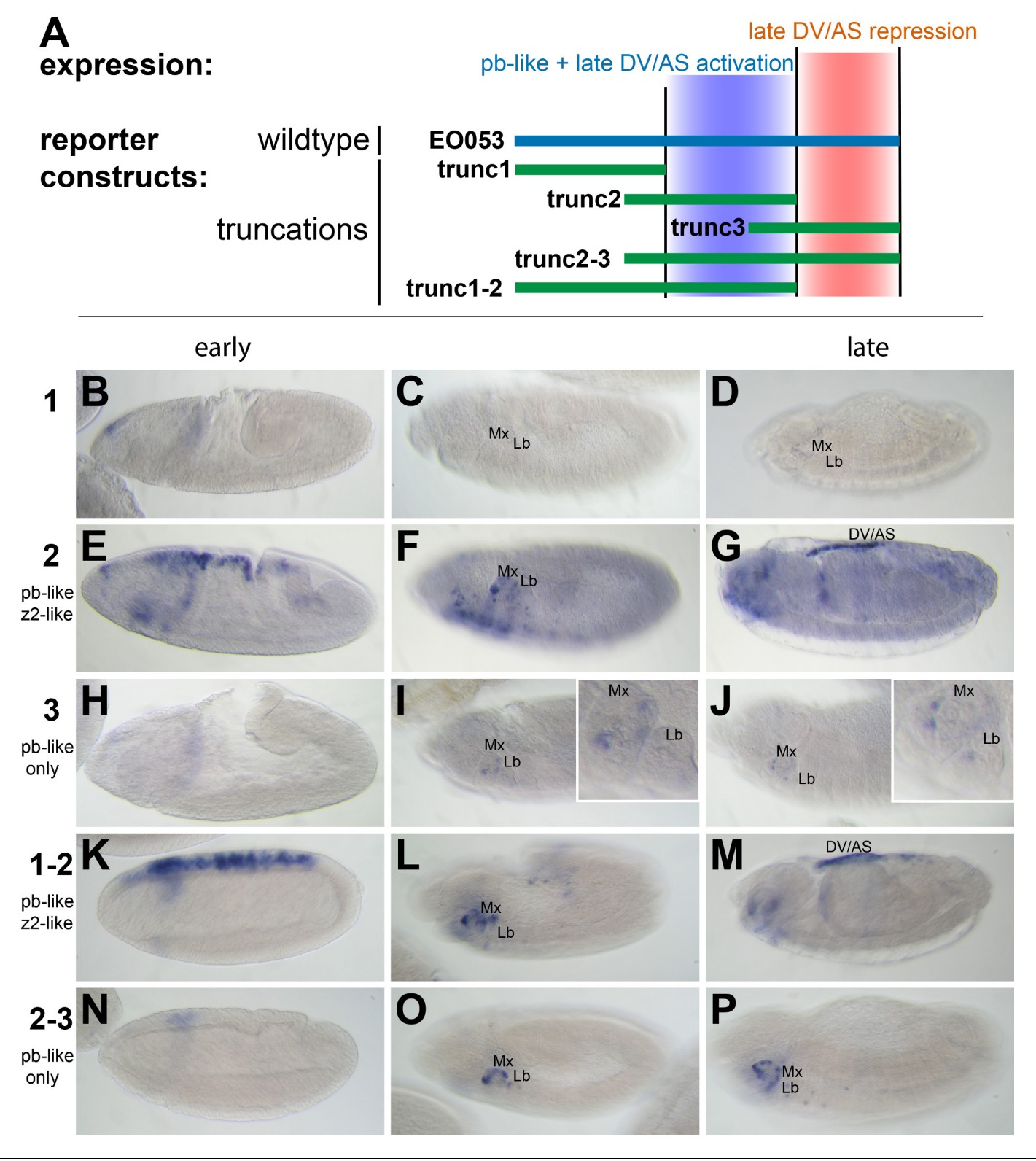

**Figure 2.** *pb*-like expression driven by EO053 can be localized to a central region of the enhancer. (A) Diagram indicating the boundaries of five truncations of EO053 (green bars) and localized expression specificities deduced from reporter assays. While *pb*-like expression can be localized to a subregion of EO053, the *zen2*-like expression cannot. 'DV/AS'=dorsal vessel/amnioserosa. (B-P) Expression of *GAL4* mRNA by in situ hybridization in transgenic reporter lines described in panel A. (B, E, H, K, N) *GAL4* expression in early embryos (stg 5–8), noting the *zen2*-like pattern in E and K only.

*Figure 2 continued on next page*

*Figure 2 continued*

E represents a rare embryo with early dorsal expression, and only during stage 6. (C, F, I, L, O) Segment labels as in *Figure 1*. *GAL4* expression in stage 10–12 embryos, noting *pb*-like expression in panels F, I, L, and O. (D, G, J, M, P) *GAL4* expression in stage 13–16 embryos. Two constructs that both contain the trunc2 region but lack the remaining 3′ portion of EO053 express ectopic *GAL4* in the DV/AS region (G, M). Insets in I and J represent zoomed-in sections highlighting the low signal in the maxillary and labial segments found with the trunc3 construct. See *Figure 1—figure supplement 1* for a diagram of these and all constructs used in this study.

Interestingly, none of the 47-nt mutations could recapitulate the strong reduction of *zen2*-like activity seen in *EO053ΔFG>GAL4*. Two mutants, FG3 and FG7, strongly reduce the *pb*-like expression pattern (*Figure 4F′–F″, J′–J″*). Each of these mutants also affects the *zen2*-like pattern, though in opposite directions: FG3 causes an ectopic anterior expansion of the *zen2*-like pattern (*Figure 4F*), while FG7 reduces *zen2*-like expression (*Figure 4J*). FG1, FG2, FG4, FG6, and FG8 also reduce expression in the *zen2*-like pattern (*Figure 4D–K*). The reduced expression seen with the FG4 mutation results in dorsal stripes (*Figure 4G*), which were also observed with the insufficient truncA-D construct (*Figure 3H*). Together, these data suggest that while the *pb*-like pattern can be effectively localized to the FG region in EO053, the elements required for *zen2*-like expression are spread much more broadly throughout EO053 and are even linked to regions necessary for the *pb*-like pattern.

## EO053 cooperates with gene-specific enhancers to direct the full expression of both *pb* and *zen2*

While the *pb*- and *zen2*-like specificities appear to be linked within the EO053 sequence, we sought to determine whether EO053 is functionally linked to either *pb* or *zen2*, or both. We thus generated via CRISPR/Cas9 a deletion of the EO053 region at the endogenous *pb* locus, designated *pb^{M2:20}*. A chromosomal deletion removing *zen2* and null for *pb*, *pb^{23}* (a.k.a. *pb^{map8}*), lacks any embryonic cuticle phenotype (*Pultz et al., 1988*). Therefore, we examined effects upon both *pb* and *zen2* mRNA accumulation (*Figure 5*). Kapoun and Kaufman have shown that *pb* mini-genes lacking large sections of the intron overlapping EO053 are capable of rescuing adult mouthparts-to-leg transformations in *pb* null flies and that the 2.1 enhancer is required for rescue in the context of these small *pb* mini-genes (*Kapoun and Kaufman, 1995a*). Thus, because the 2.1 enhancer is unaffected in *pb^{M2:20}* homozygous embryos, we were not surprised to observe detectable *pb* mRNA in these embryos (*Figure 5B,C*), as well as in labial discs from 3^{rd}-instar larvae (*Figure 5—figure supplement 1*). Kapoun and Kaufman also showed that a 10.6-kb fragment—apparently overlapping EO053 sequence—was able to drive *LacZ* expression in maxillary and, to a lesser extent, labial segments in embryos (*Kapoun and Kaufman, 1995a*). While this is strikingly similar to the EO053 expression pattern we observe, it is possible that additional *pb* enhancers outside of EO053 reside on this 10.6-kb fragment. Despite 2.1 and other potential *pb* enhancers remaining intact in *pb^{M2:20}* flies, in double-blind scoring of *pb* expression in parallel in situ hybridization experiments, we were able to observe a statistically significant reduction in *pb* mRNA accumulation in *pb^{M2:20}* embryos compared to *w^{1118}* controls (*Figure 5—figure supplement 2A–C*). We failed to validate this difference by qPCR in staged embryos, however (*Figure 5—figure supplement 2D,E*), and suspected that region 2.1 may be masking the consequence of EO053 deletion. Consistent with this hypothesis, deletion of region 2.1 alone was sufficient to cause a noticeable reduction in expression area in maxillary and labial segments in mutant embryos (*Figure 5E*, *Figure 5—figure supplements 3*, *4*) and in labial discs from 3^{rd}-instar larvae (*Figure 5—figure supplement 1*), and to cause a proboscis-to-leg transformation in adult flies (*Figure 5—figure supplement 5*) reminiscent of *pb* null mutants. The remaining *pb* mRNA detectable in Δ2.1 single mutants, however, was reduced to an even greater extent when EO053 was also deleted, as observed in maxillary and labial segments in double mutant embryos (*Figure 5F*, *Figure 5—figure supplements 3*, *4*) and in labial discs (*Figure 5—figure supplement 1*). This strong effect upon *pb* mRNA expression relative to the Δ2.1 single mutants did not appear to enhance the proboscis-to-leg transformation, however (*Figure 5—figure supplement 5*). These data suggest that indeed EO053 serves a role as a dual enhancer of *pb*, operating in addition to the 2.1 and potentially other enhancers (*Kapoun and Kaufman, 1995a*).

We similarly anticipated that if EO053 regulates *zen2* it may not act alone on this target either. Endogenous *zen2* mRNA accumulation expands to the anterior and posterior poles of blastoderm-

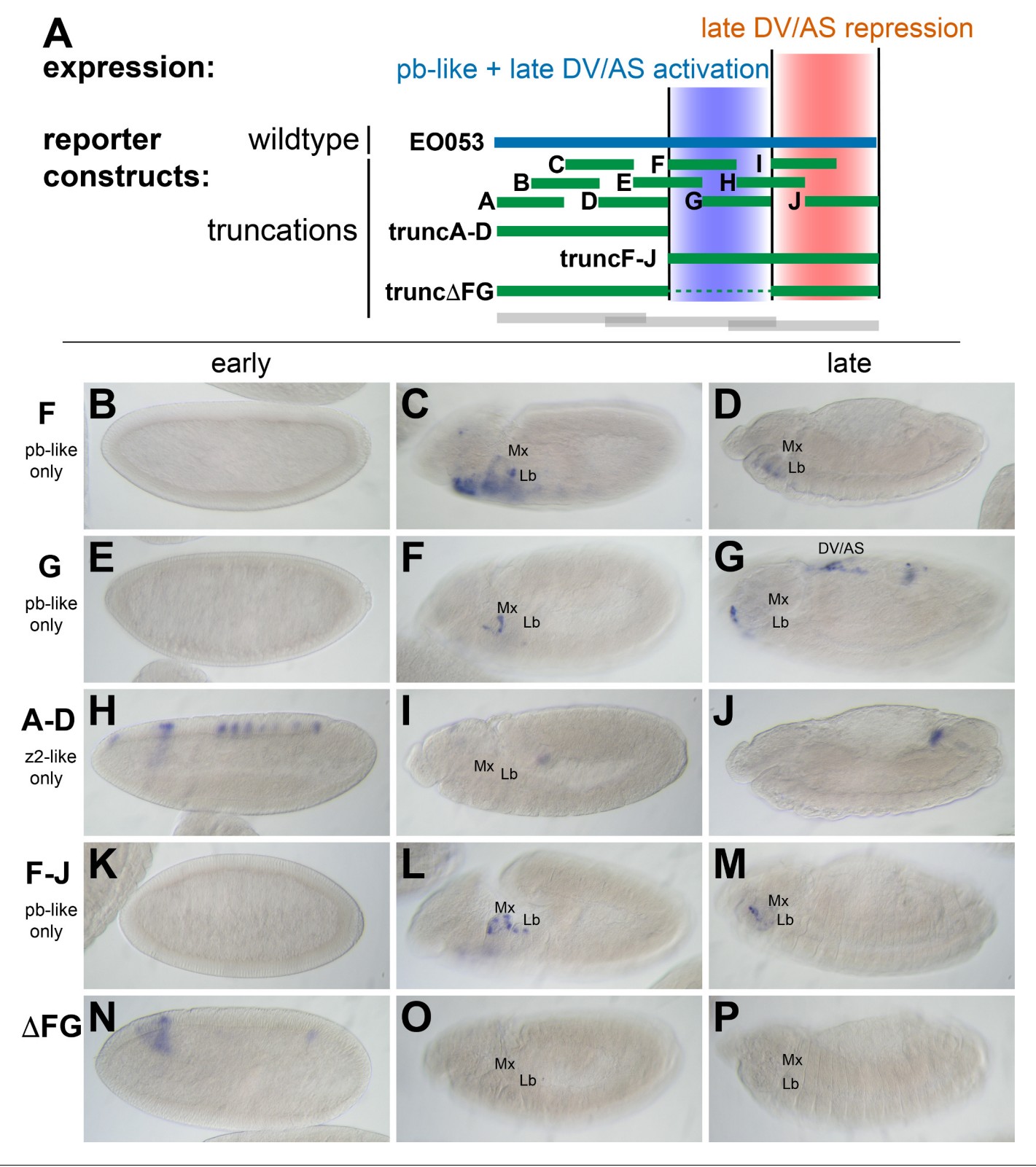

**Figure 3.** *zen2*-like expression driven by EO053 requires the central region of the enhancer. (**A**) Diagram indicating relative locations of the second set of constructs representing truncated versions of EO053 (green bars). Boundaries of the constructs shown in *Figure 2* are indicated for comparison (grey bars). (**B-P**) Expression of *GAL4* mRNA by in situ hybridization in a subset of transgenic reporter lines described in A (See *Figure 3—figure supplement 1* for images of truncA – truncJ). DV/AS: dorsal vessel/amnioserosa; segment labels as in previous figures. (**B, E, H, K, N**) *GAL4* expression in early

*Figure 3 continued on next page*

*Figure 3 continued*

embryos (stg 5–8), noting the striped *zen2*-like pattern in **H** only. (**C, F, I, L, O**) *GAL4* expression in stage 10–12 embryos, noting *pb*-like expression in panels **C, F,** and **L**. (**D, G, J, M, P**) *GAL4* expression in stage 13–16 embryos. truncG overlaps trunc2 region but lacks the remaining 3' portion of EO053 and expresses ectopic *GAL4* in the DV/AS region (**G**). (**N-P**) truncΔFG, which lacks regions **F** through **G**, fails to express *GAL4* in either *pb*- or *zen2*-like patterns. See also *Figure 1—figure supplement 1*.

The online version of this article includes the following figure supplement(s) for figure 3:

**Figure supplement 1.** Embryo images of truncA – truncJ expression patterns.

stage embryos (*Figure 5I*), a pattern that differs from expression driven by EO053, which is absent from the poles (*Figure 1B*). While the regulation of *zen2* expression has not yet been pursued, an investigation of the paralogous *zen* gene found that a reporter driven by the promoter-proximal region recapitulates endogenous *zen* gene expression (*Doyle et al., 1989*). Guided by the *zen2* inversion between *D. melanogaster* and *D. virilis* (*Figure 1—figure Supplement 1D*), we cloned the *zen2* promoter-proximal region (zen2US) and found that it is indeed capable of driving reporter expression in a pattern that recapitulates the polar expansion of endogenous *zen2* mRNA (*Figure 5G*). Intriguingly, the EO053 and zen2US patterns differ not only spatially but temporally, with zen2US driving reporter expression only until the completion of cellularization at stage 5 (*Figure 5G,H*), while EO053 is strongly active at this stage and continues to gastrulation (*Figure 1B–D*). Such a transition is also observed with endogenous *zen2* mRNA: Early expression includes expression in polar regions as well as dorsally at stage 4 (*Figure 5I*), but following cellularization the mRNA is largely detectable only in dorsal-most cells and is absent from the anterior and posterior poles (*Figure 5J*). We find that only this later and not the earlier accumulation of endogenous *zen2* mRNA requires EO053, as $pb^{M2:20}$ embryos largely fail to express *zen2* beyond cellularization (*Figure 5L,L'*). Thus, EO053 appears to have dual roles in *Drosophila* embryogenesis, assisting other enhancers in early stages with *zen2* expression and then with *pb* expression during later morphogenetic events (Figure 7).

### *pb* and *zen* genes remain syntenic despite the change in *zen* expression

The *zen*, *zen2*, and *bicoid* (*bcd*) genes in *Drosophila* are derivatives of the ancestral *Hox3* ortholog in basal arthropods, and have diverged in expression and function from the ancient homeotic role. Why, then, do they remain at their ancestral genomic location within the Hox complex? Splits and inversions within the Hox complex are common in Schizophoran flies, suggesting loosened constraints on colinearity (*Negre and Ruiz, 2007*; *Von Allmen et al., 1996*). The sharing of regulatory elements among members of the Hox complex has been a model to explain the persistent linkage of Hox genes in metazoan genomes (*Spitz et al., 2003*; *Sharpe et al., 1998*), and the function of EO053 provides direct support for maintenance of an ancestral regulatory linkage as a contributing factor to persistence of a *pb-zen* linkage. While it is challenging to trace EO053 itself across evolution, patterns of synteny between *pb* and *zen* following the functional transition offer an opportunity to test such a model. Specifically, any translocation of a *zen* ortholog away from the *pb* ortholog would presumably not be favored if one or more regulatory elements are shared between the two genes. Indeed, examining available genomic scaffolds across 80 different Arthropods, we were unable to detect any translocation event that breaks the synteny between *pb/Hox2* and *zen/Hox3* (*Figure 6—figure supplements 1–9*). Furthermore, among the 66 species examined that evolved following the *Hox3/zen* divergence, only three species—all members of the Formicoidea—exhibit a change in synteny: only via the loss of the *zen* coding sequence (*Figure 6—figure supplements 1* and *5*). In contrast, 3/14 species examined that predate the *Hox3/zen* divergence exhibit loss of *Hox3* or *pb*—representing Crustacea, Myriapoda, and Chelicerata (*Figure 6—figure supplements 1, 8* and *9*; *Chipman et al., 2014*; *Grbić et al., 2011*; *Pace et al., 2016*; *Kim et al., 2016*; *Kenny et al., 2014*).

### An EO053 motif important for both *pb*- and *zen*-like expression exhibits patterns of conservation within various clades

Examining patterns of regulatory sequence conservation is a complementary approach to exploring the model of ancient, shared regulation as an explanation of the persistent *pb/zen* linkage.

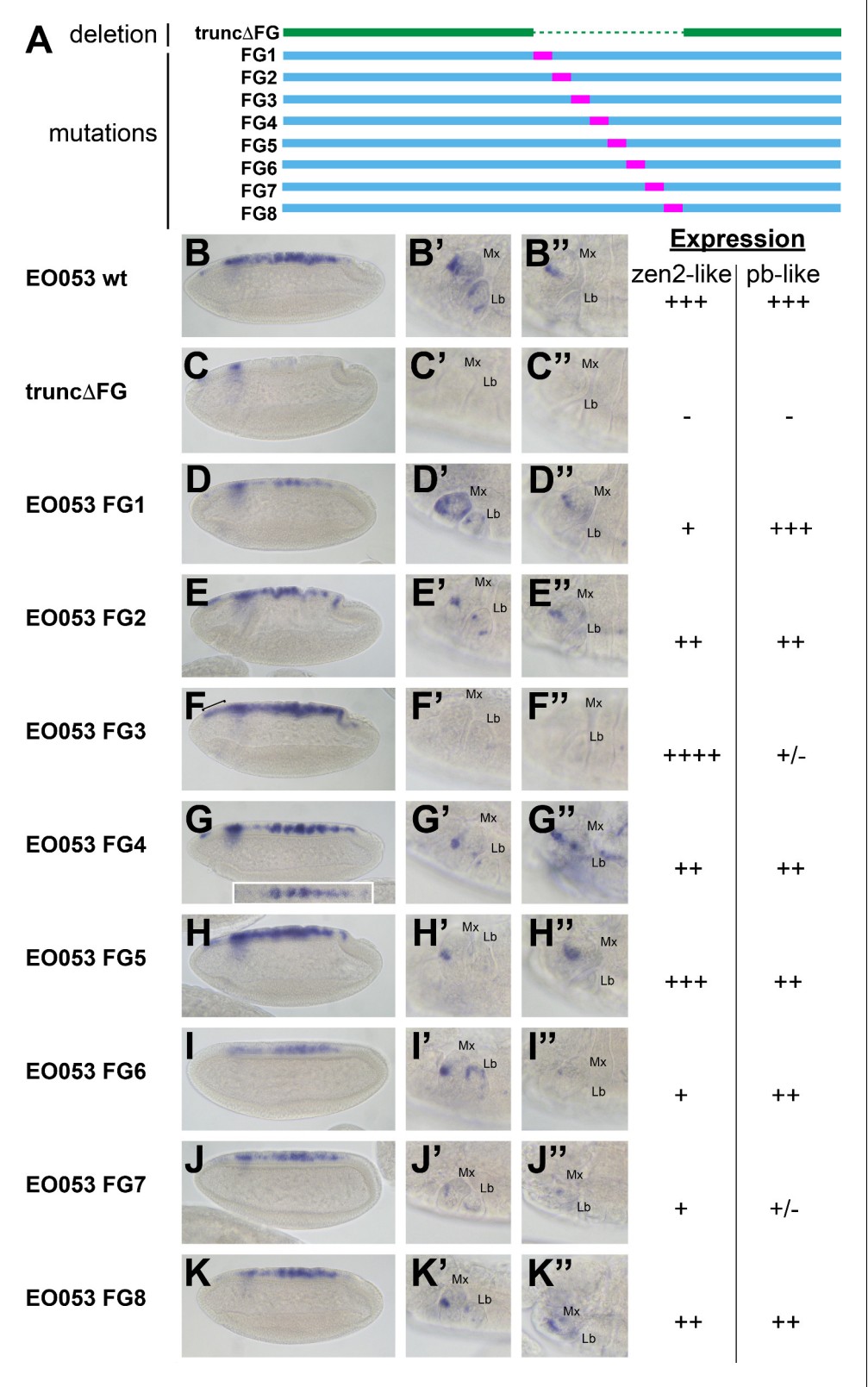

**Figure 4.** Mutation of specific nucleotide segments in the FG region of EO053 can affect either *pb*-like or *zen2*-like expression. (A) Diagram of a series of 47-nt non-complementary transversion mutants generated within the FG region (FG1 – FG8, blue, with mutated segments shown in pink), and the same region deleted in the truncΔFG construct (green). (B-K) *GAL4* mRNA expression in early (stage 4–6) embryos. *zen2*-like expression is absent in truncΔFG (C); reduced in FG1 (D), FG2 (E), FG4 (G), FG6 (I), FG7 (J), and FG8 (K); and expanded anteriorly in FG3 (F: bracket). Inset in G is a dorsal view

*Figure 4 continued on next page*

*Figure 4 continued*

of an embryo exemplifying the pseudo-stripe pattern of *GAL4* expression along the anteroposterior axis driven by the FG4 mutant reporter. (B'-K'') *GAL4* mRNA expression in maxillary and labial segments of stage 10–12 embryos (B'–K') and stage 13–16 embryos (B''–K''). Segment labels as in previous figures. *pb*-like expression is absent in truncΔFG (C', C'') and strongly reduced in FG3 (F', F'') and FG7 (J', J''). Qualitative scoring of reporter strength is represented to the right of the images for each line. See also *Figure 1—figure supplement 1*.

Sequence conservation makes possible the identification of EO053 throughout the Schizophora (*Figure 6A,B*). In particular, 33/36 nt of the region containing the 5' 12nt of FG4 and the 3' half of FG3, the latter of which we have shown to be required for the proper expression of both *pb*- and *zen2*-like specificities (*Figure 4F–F''*), are identical between *D. melanogaster* and *Ceratitis capitata* (*Figure 6C*). Outside of the Brachycera it is challenging to identify orthologous regulatory regions. Comparing *D. melanogaster* EO053 and *pb* intronic sequence from the mosquito *Anopheles gambiae* identified a 12-nt sequence from within the Schizophora 36-nt span (ATCATTAATCAT, henceforth referred to as 'the EO053 motif', in green in *Figure 6B,C*) that is also found in the *Anopheles* intron (*Figure 6—figure supplement 2*). This sequence is similar to others that have been shown to be bound by Exd/Hox dimers (*Bergson and McGinnis, 1990*; *Chan et al., 1997*; *Regulski et al., 1991*; *Rusch and Kaufman, 2000*; *Zeng et al., 1994*), and thus represents a plausible candidate for an ancient motif with regulatory function. We find that this motif is important for EO053 function, as a TTAA>GGCC mutation to abrogate Hox binding dramatically reduces GAL4 expression in both the *zen2*-like and *pb*-like specificities (*Figure 6D–F'*). The deepest we are able to identify a region orthologous to EO053 outside of Schizophora is in the assassin fly *Proctacanthus coquilletti* (Brachycera; Orthorrapha). This species contains a variant EO053 motif (ATCATAAATCAT) that could still mediate an Exd/Hox interaction (*Slattery et al., 2011*). Given the functional importance of this motif for both aspects of EO053 expression and its conservation within Brachycera, we chose to examine the 80 Arthropod *Hox2/3* regions for patterns consistent with an ancient regulatory function for this or similar sequences.

65/66 species that arose following the *Hox3/zen* divergence contain instances of the EO053 motif within the large introns between the YPWM- and homeodomain-encoding exons of *pb* (*Figure 6—figure supplements 2–9*). The outlier *Locusta migratoria* scaffold containing the homeodomain coding sequence lacks upstream motif instances but also lacks the upstream exon necessary to confirm motif absence (*Figure 6—figure supplement 7*). In species that arose prior to the *Hox3/zen* divergence, the *Ixodes scapularis* (Chelicerata) (*Figure 6—figure supplement 9*), *Daphnia pulex* (Crustacea) (*Figure 6—figure supplement 8*), and *Orchesella cincta* (Hexapoda) (*Figure 6—figure supplement 7*) *pb* orthologs lack intron motif instances. Despite a lack of direct evidence for which, if any, motifs in the *Hox2/3* regions are functional outside of Schizophora, we nevertheless examined these genomic intervals for patterns of conservation.

We identified in several major Arthropod clades conserved instances of the EO053 motif, based upon relative location and flanking sequences (*Figure 6—figure supplements 2–9*, *Supplementary file 1*). Lepidoptera contain a conserved motif instance between *pb* and *zen2* (Motif 2 in *Figure 6—figure supplement 3*, *Supplementary file 1*). Coleoptera contain a conserved mismatch upstream of the *pb* promoter (Motif 3 in *Figure 6—figure supplement 4*, *Supplementary file 1*), and several species contain conserved motifs within the *pb* intron (Motif 4) and upstream of the duplicated *zen* genes, including on an isolated scaffold harboring a fourth *zen* paralog in the Asian long-horned beetle *A. glabripennis* (Motif 5 in *Figure 6—figure supplement 4*, *Supplementary file 1*). Within Hymenoptera, the two basal Tenthredinoidea species have five conserved motif mismatches (Motifs 6–10 in *Figure 6—figure supplement 5*, *Supplementary file 1*). Outside the Tenthredinoidea, all remaining species conserve a mismatch upstream of *zen* (Motif 11 in *Figure 6—figure supplement 5*, *Supplementary file 1*). Of particular note, this sequence is still present in the three Formicoidea that have lost the *zen* coding sequence. The Formicoidea also contain an intron motif specific to this clade (Motif 13 in *Figure 6—figure supplement 5*, *Supplementary file 1*), as well as another intron motif also conserved throughout the Aculeata (Motif 12 in *Figure 6—figure supplement 5*, *Supplementary file 1*). Several of the Hemiptera examined (excluding the Sternorrhyncha species *A. pisum*, *D. citri*, and *B. tabaci*) contain a conserved motif mismatch downstream of *zen* (Motif 14 in *Figure 6—figure supplement 6*, *Supplementary file 1*), and the two Dictyoptera species contain a conserved motif mismatch

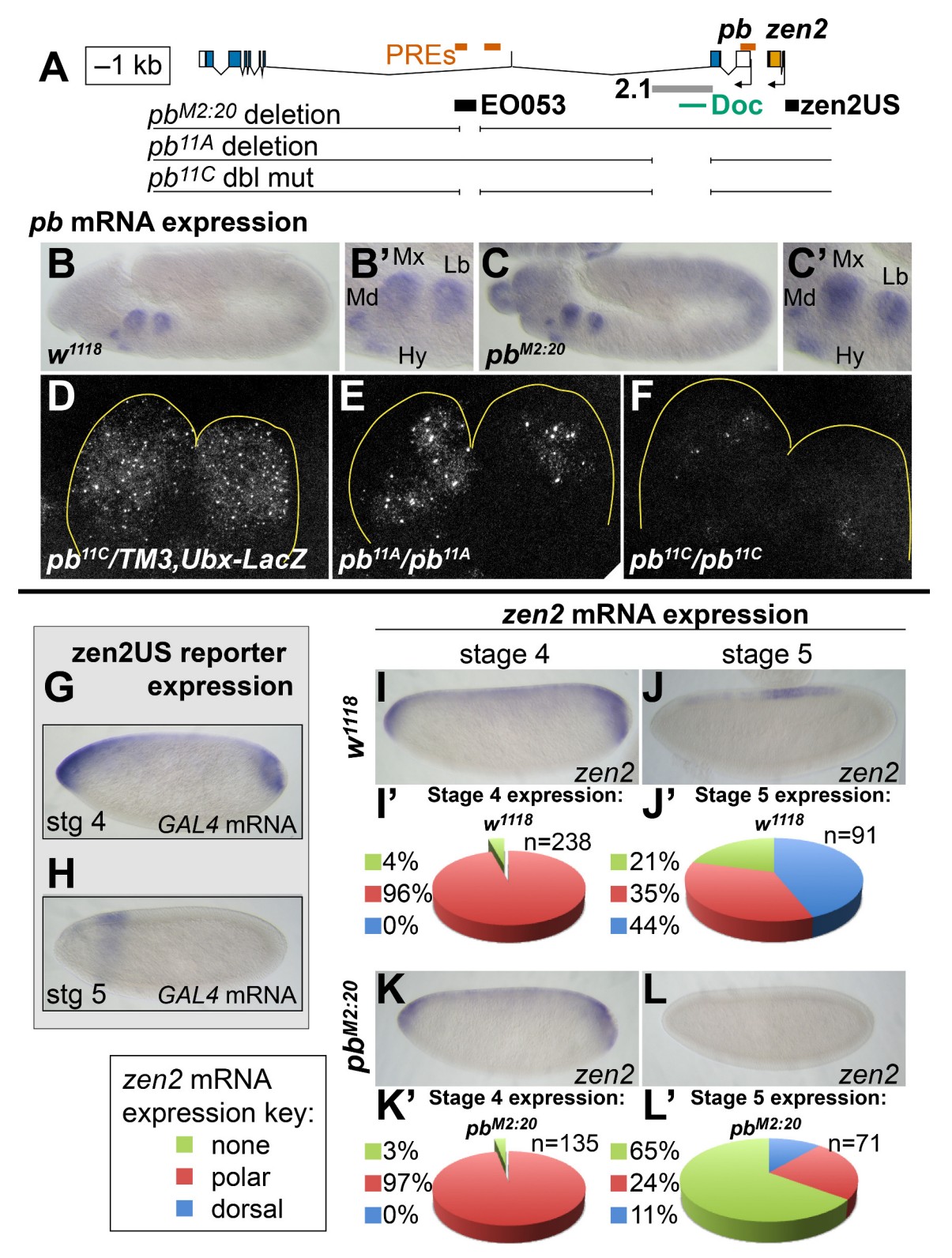

**Figure 5.** EO053 cooperates with other enhancers to regulate mRNA accumulation from both *pb* and *zen2*. (**A**) Diagram of the *pb-zen2* region, noting the locations of putative Polycomb Response Elements ('PREs', red; see Discussion) (**Nègre et al., 2011**); EO053 and zen2US enhancer regions (black) and the *pb* 2.1 regulatory region (grey) (**Kapoun and Kaufman, 1995a**); and a Doc type transposon (**Vaury et al., 1994**) in the 5' end of *pb* intron 2. Below the diagram of the genomic region are shown CRISPR/Cas9-generated deletions overlapping EO053 only (*pb^{M2:20}*), the 2.1 enhancer only (*pb^{11A}*

*Figure 5 continued on next page*

Figure 5 continued

or the identical *pb^11D* seen in *Figure 5—figure supplement 3* and *Figure 5—figure supplement 1*), and both EO053 and 2.1 enhancers (*pb^11C* or the identical *pb^11E* seen in *Figure 5—figure supplement 3* and *Figure 5—figure supplement 1*). (B-F) Effects of enhancer deletion on *pb* expression. (B) *pb* mRNA expression in a *w^1118* embryo at stage 11–12. (B′) Zoom-in of the *pb* in situ signal in the mandibular (Md), maxillary (Mx), labial (Lb) segments, and hypopharyngeal lobe (Hy). (C) *pb* mRNA expression in a *pb^M2:20* embryo at stage 11–12. (C′) Zoom-in of the *pb* in situ signal, with labeling as in B′. See also *Figure 5—figure supplement 2*. (D-F) Confocal maximum projection of *pb* mRNA detected through fluorescent in situ hybridization (FISH) in the maxillary and labial segments of embryos of the indicated genotypes. (D) *pb^11C*/TM3,Ubx-LacZ stage 11–12 embryo. (E) *pb^11A*/*pb^11A* stage 11–12 embryo, noting dramatically reduced signal area relative to D. (F) *pb^11C*/*pb^11C* stage 11–12 embryo exhibiting signal area reduced relative to D and E. See also *Figure 5—figure supplement 3* – 5. (G, H) Expression of *GAL4* directed by the reporter zen2US. Embryos containing *zen2US>GAL4* have detectable *GAL4* mRNA expression at stage 4 (G) and lack *GAL4* expression during stage 5 (H). (I-L′) Effect of the *pb^M2:20* deletion on *zen2* expression. *zen2* mRNA expression at either stage 4 (I,K) or stage 5 (J,L) in *w^1118* embryos (I,J) or *pb^M2:20* embryos (K,L). (I′,J′,K′,L′) Pie-chart representation of *zen2* mRNA expression pattern as resembling zen2US (polar, red), EO053 (dorsal, blue), or absent (none, green).

The online version of this article includes the following source data and figure supplement(s) for figure 5:

**Source data 1.** Scoring Data for *pb^M2:20* *pb* and *zen2* in situ phenotypes (*Figure 5* and *Figure 5—figure supplement 2*).
**Figure supplement 1.** Fluorescent detection of *pb* mRNA in 3^rd-instar labial discs.
**Figure supplement 2.** Quantification of *pb* expression in *pb^M2:20* mutant embryos.
**Figure supplement 2—source data 1.** Raw qPCR data and analysis.
**Figure supplement 3.** Region 2.1 and EO053 cooperate to drive *pb* expression.
**Figure supplement 4.** Fluorescent detection of *pb* mRNA in the maxillary and labial lobes in region 2.1 single deletions and 2.1, EO053 double deletions.
**Figure supplement 5.** Deletion of region 2.1 is sufficient to cause a proboscis-to-leg transformation.

upstream of *zen* (Motif 15 in *Figure 6—figure supplement 7*, *Supplementary file 1*). Within the Chelicerata, gene duplication and loss present interesting opportunities to examine motif instances. The multiple rounds of whole-genome duplication in the basal *Limulus polyphemus* led to three copies each of *pb* and *Hox3*, two of which have confirmed synteny. All three *Hox3* paralogs have a motif mismatch downstream (Motif 16 in *Figure 6—figure supplement 9*, *Supplementary file 1*). At a similar position downstream of one of the *pb* paralogs is the same 12-nt motif, with a variant site at the same position downstream of a second *pb*. We found another curious parallel between one adjacent *Limulus pb* and *Hox3*: the same 12-mer mismatch motif is found in the introns of each gene, though upstream of the putative *Hox3* coding sequence, at a similar distance from a separate motif instance (Motif 17 in *Figure 6—figure supplement 9*, *Supplementary file 1*). This same motif was also found upstream of the coding sequence of the third *Limulus Hox3*, again at a similar distance from another motif instance. We also found instances of these motifs in Arachnida, with a *Hox3* ortholog in *Centruroides exilicauda*, *Parasteatoda tepidariorum*, and *Stegodyphus mimosarium* as well as a *pb* ortholog in *Centruroides* and *Parasteatoda* having the downstream motif (Motif 16); we also found a 12-mer matching the intron motif (Motif 17) in one of the *pb* paralogs in *Stegodyphus* (*Figure 6—figure supplement 9*, *Supplementary file 1*). By contrast, we observed largely no conservation of 40 randomly generated 12mers (also allowing for a 1-bp mismatch) even when examining the Schizophoran regions, for example, with a single motif instance occurring every 772,189 bp on average (*Supplementary file 2*). Together our analysis indicates that while functional assumptions are limited to the Schizophora, sequences resembling the EO053 motif exhibit patterns of conservation across Arthropod clades within the *Hox2/Hox3* genomic region, particularly within clades emerging after the *Hox3/zen* divergence.

## Discussion

We have shown that the EO053 enhancer exhibits inherent regulatory complexity in two critical ways. First, it encodes more than one specificity, manifested by its ability to drive reporter gene expression in two distinct temporal and spatial patterns: the *zen2*-like dorsal expression in stage 5–7 embryos and the *pb*-like maxillary and labial segment expression beginning in stage 10. Second, each of these specificities is functionally linked to regulation of a separate target promoter. As such, this region serves as a curious contrast to existing models of complex regulatory output derived from examples of multiple independent enhancers working additively on a single target gene (*Simonet et al., 1991*; *MacNeill et al., 2000*; *Harding et al., 1989*), a single enhancer directing multiple specificities on a single target gene (*Betancur et al., 2011*; *Nagy et al., 2018*; *Preger-*

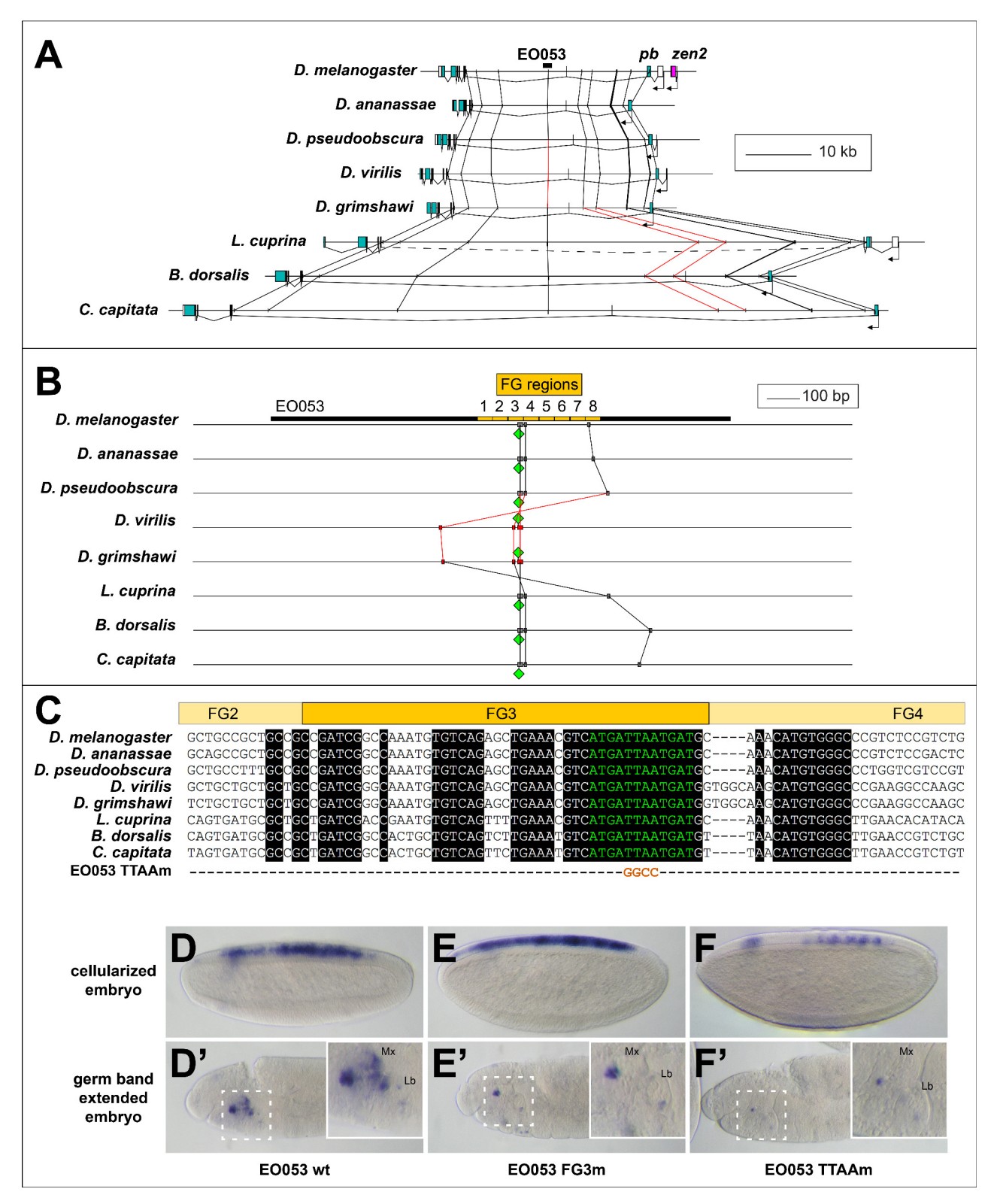

**Figure 6.** Conservation of EO053 sequences within the Schizophora. (**A**) Gene diagrams of *pb* from select Schizophoran flies with available genome sequence data. Coding exons of *pb* in each species are colored blue-green, based upon existing genome annotations, and the locations of EO053 and *zen2* in *D. melanogaster* are also noted. Vertical lines between species diagrams connect 14 bp or greater identical sequence blocks present in all eight species. Red lines (e.g., connected to the corresponding EO053 regions in *D. virilis* and *D. grimshawi*) represent sequences inverted relative to *D.*

*Figure 6 continued on next page*

eLife Research Communication

Developmental Biology | Genetics and Genomics

*Figure 6 continued*

*melanogaster* (see also *Figure 1—figure supplement 1*). Dashed line in *L. cuprina* diagram joins two separate coding regions annotated as belonging to *pb*, due to the presence of coding sequences for a YPWM motif (right-most exons) and a homeodomain (left-most exons). (B) Diagram of EO053 sequence conservation within select Schizophoran flies. *D. melanogaster* EO053 span is indicated by the thick black line and yellow boxes represent the boundaries of FG regions mutated in *Figure 4*. Grey or red boxes connected between species represent 8 bp or greater identical sequence blocks present in all eight species. Green diamonds denote the location and orientation of the conserved 'EO053 motif' sequence shown in green in panel **C**. (C) Alignment of the region including FG3 from select Schizophoran flies, indicating additional sequences conserved in this region in these species (see also *Figure 6—figure supplements 1–9*). Line below the alignment indicates the nucleotides mutated in the 'TTAAm' reporter construct shown in F. (D-F') GAL4 expression in embryos carrying mutant FG3 region reporter constructs in either early embryos (**D–F**) or germ band extended embryos (**D'–F'**). (D, D') Wildtype EO053 reporter. (E, E') Noncomplementary transversion FG3 mutant reporter. (F, F') 'TTAAm' reporter, mutating the four nucleotides indicated in **C**. Insets in **D'-F'** show higher-magnification images of the maxillary and labial segments.

The online version of this article includes the following source data and figure supplement(s) for figure 6:

**Figure supplement 1.** Synteny of *pb* and *zen2* orthologs across Arthropoda.
**Figure supplement 1—source data 1.** Table of acquired genomic scaffold accession numbers.
**Figure supplement 2.** Instances of motifs similar to the EO053 conserved Hox-like motif in the *pb* region across Arthropods.
**Figure supplement 3.** A motif upstream of *pb* (2) is conserved in Lepidoptera.
**Figure supplement 4.** Several motif instances (3, 4, 5) show conservation within the Coleoptera.
**Figure supplement 5.** Conservation of several motifs (6-13) within Hymenoptera.
**Figure supplement 6.** A single motif instance (14) appears conserved within some of the Hemiptera, excluding Sternorrhyncha.
**Figure supplement 7.** Diverse basal Hexapods include a motif (15) found in both the termite and cockroach.
**Figure supplement 8.** Crustacea and Myriapoda lack motif conservation and exhibit loss of *pb* and/or *Hox3* orthologs.
**Figure supplement 9.** Some Chelicerates contain similar motifs (16, 17) near the duplicated *pb*/*Hox3* genes.

*Ben Noon et al., 2018*), or control regions conferring common expression patterns upon multiple local target genes (*Choi and Engel, 1988*; *Deschamps, 2007*; *Foley et al., 1994*; *Lehoczky et al., 2004*; *Sharpe et al., 1998*; *Spitz et al., 2003*; *Tsai et al., 2016*; *Jones et al., 1995*; *Tsujimura et al., 2007*; *Mohrs et al., 2001*; *Cheng et al., 2014*).

## Serving separate promoters

Perhaps the most curious feature of EO053 is its requirement by distinct genes for the reliability (*pb*) or temporal progression (*zen2*) of their expression. Given its intronic location, we suggest that this regulatory arrangement would be mediated by a looping interaction between EO053 and each target promoter (*Levine et al., 2014*; *Matharu and Ahituv, 2015*). Such interactions are likely permitted by the distinct temporal activation profiles of each target gene, allowing the enhancer to separately engage only a single active promoter at a time (*Figure 7*). In addition, we have gained insight into the temporal dynamics with which EO053 operates on the *zen2* locus, whereby the initial activation of *zen2* expression is mediated by the promoter-proximal zen2US segment and then switches to control by EO053.

The regulatory interactions between EO053 and both *pb* and *zen2* are likely influenced by the overall regulatory architecture of the *Antp* Complex. Recent high-resolution analyses of topologically associated domains (TADs) suggest that *pb*, *zen2*, *zen*, and *bcd* all reside in a single TAD (*Eagen et al., 2017*; *Stadler et al., 2017*), potentially biasing regulatory activities between these loci and separate from *Dfd*, which is a regulatory island (*Stadler et al., 2017*). The three-dimensional architecture facilitating interactions between EO053 and its target promoters is likely mediated by Polycomb Response Elements (PREs) that have been mapped within the *pb* locus (*Figure 5A*; *Kapoun and Kaufman, 1995b*; *Nègre et al., 2011*; *Stadler et al., 2017*) and exhibit chromosomal pairing experimentally (*Kapoun and Kaufman, 1995b*). The observed establishment of Polycomb bodies in the nucleus at stage 5 (*Cheutin and Cavalli, 2012*) also correlates well with the temporal shift in regulation of *zen2* from promoter-proximal to distal regulatory regions.

## Distinct temporal and spatial specificities

We have shown that the *pb*-like and *zen2*-like specificities overlap within the FG region of EO053. While the *pb*-like expression is largely restricted to this region, the *zen2*-like expression appears to be much more broadly extended throughout EO053. Moreover, within the FG region, mutation of FG3 or FG7 affects both specificities, suggesting the specificities may have one or more motifs in

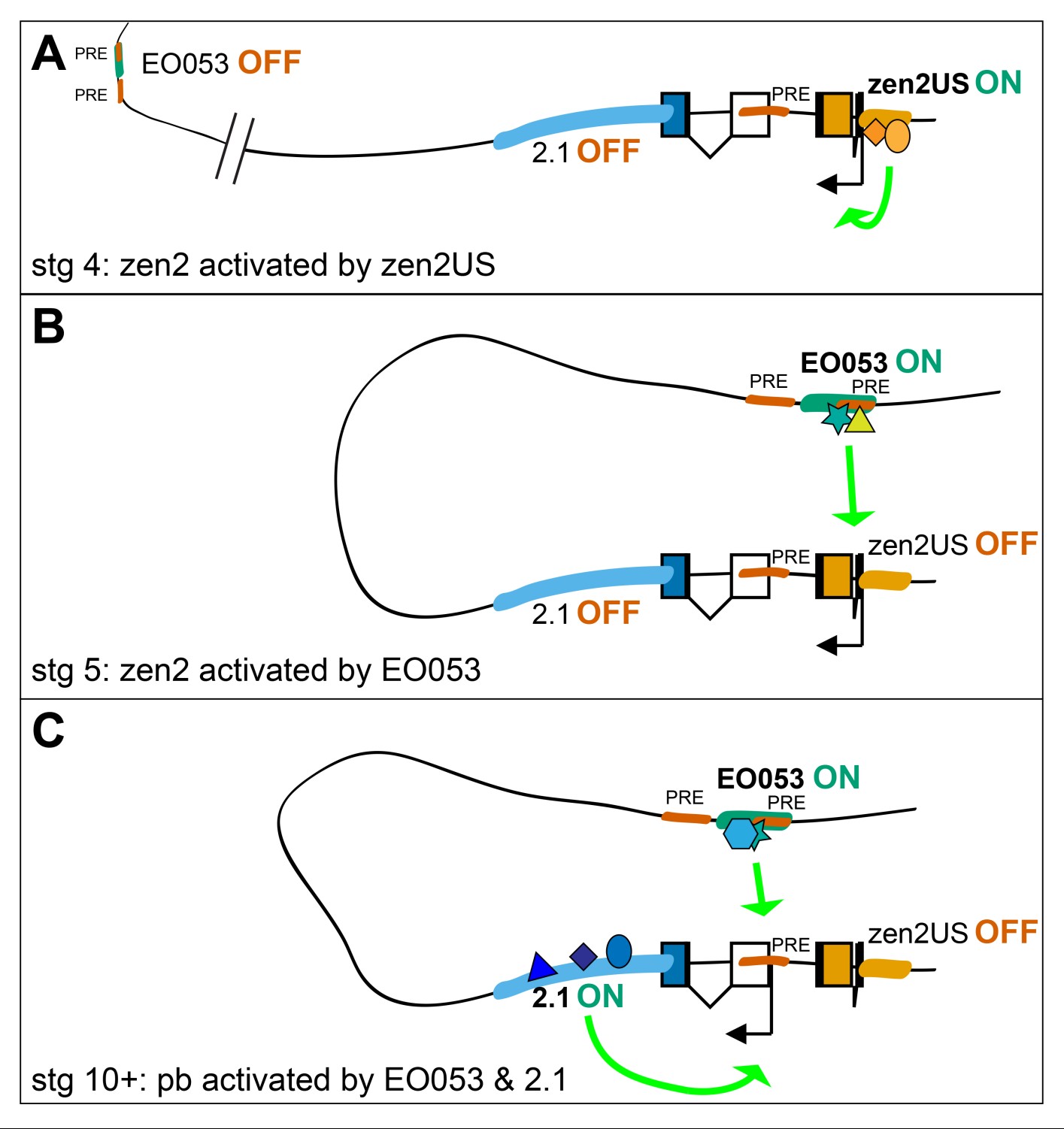

**Figure 7.** Possible model for EO053 regulation of both *zen2* and *pb*. (A-C) Diagram of the dynamic activities of EO053 during development. Black arrows indicate active transcription of either *zen2* (orange exons) or *pb* (white and blue exons), and green arrows signify active enhancers regulating transcription of either promoter. (**A**) At stage 4 in the dorsal blastoderm and at both anterior and posterior poles, *zen2* expression is initiated by the upstream enhancer, zen2US. (**B**) As zen2US loses activity in stage 5, expression of *zen2* instead becomes dependent upon EO053 in the dorsal blastoderm, potentially mediated by chromatin looping. (**C**) Later, in the developing head primordium, EO053 assists region 2.1 in directing *pb* expression, which may be mediated by interactions involving factors bound to nearby PREs (red in **A-C**; see also gene diagram in *Figure 5*). The online version of this article includes the following figure supplement(s) for figure 7:

*Figure 7 continued on next page*

*Figure 7 continued*

**Figure supplement 1.** Modifying the reporter promoter does not affect expression pattern driven by EO053.

common in their regulatory logic. Such a motif could be bound by the same transcription factor in both settings or related factors with different temporal and/or spatial profiles. We raise the possibility that the conserved EO053 motif may represent such a site. First, its pattern of conservation and location within the functionally important FG3 region suggests that this sequence itself may be required. Second, we show that mutating the core Hox motif affects both the *pb*-like and *zen2*-like expression in the context of EO053 (*Figure 6*). Similar sequences have been identified as functionally relevant to the expression of *Dfd* (*Chan et al., 1997*; *Zeng et al., 1994*; *Bergson and McGinnis, 1990*; *Regulski et al., 1991*; *Lou et al., 1995*) and *pb* itself (*Rusch and Kaufman, 2000*). Both of these examples involve regulation by Dfd, and we suspect the *pb*-like expression mediated by this motif would likely also involve Dfd. We also notice instances of this motif upstream of the *Dfd* orthologs themselves in many of the species we have analyzed (*Figure 6—figure supplements 2–9*), suggesting that *Dfd* auto-regulation within the arthropods may be ancient. Given the similar binding specificities of Hox proteins (*Noyes et al., 2008*; *Berger et al., 2008*), this site could be utilized by one or more of these proteins, even operating as a promiscuous auto-regulatory enhancer. This is consistent with the demonstrated role of EO053 in the temporal dynamics of *zen2* expression where its activity is preceded by zen2US activity, and also with our observation that region 2.1 deletion significantly affects the early *pb* expression in maxillary and labial segments (*Figure 5—figure supplement 3E*). We expect activation and specificity to also involve key regulators binding non-Hox motifs, which again could participate in one or both specificities.

Why does neither *pb* nor *zen2* mRNA expression reflect the complete regulatory capacity of EO053? A common feature of many enhancers is their ability to interact reliably with a heterologous promoter to drive reporter gene expression in a manner that not only largely recapitulates specificity encoded by the enhancer, but also represents a subset of the expression pattern of the endogenous gene. EO053, however, produces a pattern that would be considered an ectopic specificity relative to the pattern of either target gene (i.e. if EO053 were only a *pb* enhancer the blastoderm spatio-temporal activity is ectopic and the mouthpart activity does not recapitulate *zen2* expression). Promoter selectivity/interpretation is a likely model to explain the different transcriptional outputs of the separate promoters that utilize EO053. The collection of core promoter elements at an individual promoter can bias promoter-enhancer compatibility (*Butler and Kadonaga, 2001*), and the *pb* promoter itself is required for expression driven by certain enhancers (*Kapoun and Kaufman, 1995a*). Replacing the promiscuous synthetic core promoter (*Pfeiffer et al., 2008*) in the pBPGUw vector used here (TATA box present) with the minimal core promoter from either *pb* or *bcd* (TATA box absent) or replacing the initiator sequence with that of *zen2* had no effect on reporter gene expression (*Supplementary file 3*, *Figure 7—figure supplement 1*), which may suggest that promoter interpretation may require additional promoter-proximal elements to mediate the appropriate spatial/temporal output. It is additionally possible that an EO053-proximal element may facilitate appropriate promoter targeting and output (*Chen et al., 2005*; *Zhou and Levine, 1999*), or another separate region (i.e. dominant repressor) may be influencing output (*Perry et al., 2011*).

## Evolution and maintenance

The stable colinearity of vertebrate Hox genes has been attributed to sharing of regulatory elements (*Gould et al., 1997*; *Gérard et al., 1996*; *Sharpe et al., 1998*). Arthropods may not share this paradigm across the complete complex, as suggested by the occurrence of rearrangements (*Pace et al., 2016*; *Faddeeva-Vakhrusheva et al., 2017*; *Wu et al., 2017*), inversions (*Negre and Ruiz, 2007*), gene loss (*Chipman et al., 2014*; *Grbić et al., 2011*), regulatory independence (*Gellon and McGinnis, 1998*; *Shippy et al., 2008*), and local chromatin organization (*Eagen et al., 2017*; *Stadler et al., 2017*). We provide evidence here for the possibility of limited shared regulation within an Arthropod Hox complex, between a true Hox (*pb*) and a neighboring Hox-derived gene (*zen2*) both sharing EO053. While we do not know how deeply this regulatory relationship extends, the organization of Hox complexes across the phylum exhibits features consistent with shared regulation. The strong synteny of *pb* and *zen* and reduced gene loss (a subset of ant genomes being the exception to date)

suggests shared regulation may have existed from the point of *Hox3/zen* divergence, if not earlier. The most likely scenario based upon the available data would be a *pb-Hox3* shared duplicate enhancer that drives a pattern of expression common to both genes. Acquisition of a novel expression specificity by the shared enhancer would then be buffered by the additional independent regulatory sequences of each gene. Alternatively, the change in spatial/temporal expression of the shared enhancer might also impart selective pressure alleviated by differential interpretation of the enhancer by each gene, and/or functional divergence. Regardless of the mechanism, EO053 serves as an unusual example of an enhancer maintaining a promiscuous relationship with two distinct gene promoters even in the context of disparately evolving expression specificities.

# Materials and methods

## Key resources table

| Reagent type (species) or resource | Designation | Source or reference | Identifiers | Additional information |
|---|---|---|---|---|
| Gene (*Drosophila melanogaster*) | pb | | FLYB: FBgn0051481 | |
| Gene (*Drosophila melanogaster*) | zen2 | | FLYB: FBgn0004054 | |
| Gene *Drosophila virilis*) | pb | | FLYB: FBgn0211025 | |
| Genetic reagent (*D. melanogaster*) | TM3, Ubx-LacZ.w+ | Bloomington *Drosophila* Stock Center | BDSC:4432; FBti0002628; RRID:BDSC_4432) | FlyBase symbol: Dmel\P{Ubx-lacZ.w$^+$}TM3 |
| Genetic reagent (*D. melanogaster* | pb$^{M2:20}$ | This study | | EO053 deletion mutant |
| Genetic reagent (*D. melanogaster* | pb$^{11A}$ | This study | | 2.1 enhancer deletion mutant |
| Genetic reagent (*D. melanogaster* | pb$^{11C}$ | This study | | EO053/2.1 enhancer double deletion |
| Genetic reagent (*D. melanogaster* | pb$^{11D}$ | This study | | 2.1 enhancer deletion mutant |
| Genetic reagent (*D. melanogaster* | pb$^{11E}$ | This study | | EO053/2.1 enhancer double deletion |
| Antibody | anti-digoxygenin (sheep polyclonal) | Sigma Aldrich | Cat. No. 11 333 089 001 | 1:500 dilution |
| Antibody | anti-digoxygenin-AP Fab fragments (sheep polyclonal) | Sigma Aldrich | Cat# 11093274910 | 1:2000 dilution |
| Antibody | anti-biotin (mouse) | Roche | Cat. #1 297 597 | 1:500 dilution |
| Antibody | Donkey anti-sheep Alexa-488 | ThermoFisher | Catalog # A-11015 | 1:500 dilution |
| Antibody | Donkey anti-mouse Alexa-555 | ThermoFisher | Catalog # A-31570 | 1:500 dilution |
| Recombinant DNA reagent | DR274 (plasmid) | Addgene | RRID:Addgene_42250 | T7 guide RNA expression |
| Recombinant DNA reagent | MLM3613 (plasmid) | Addgene | RRID:Addgene_42251 | T7 Cas9 expression vector |
| Recombinant DNA reagent | pU6-BbsI-chiRNA (plasmid) | Addgene | RRID:Addgene_45946 | Guide RNA cloning vector for *Drosophila* injection |
| Recombinant DNA reagent | pGEM-T (plasmid) | Promega | Cat # A3600 | Cloning vector |
| Recombinant DNA reagent | pBPGUw (plasmid) | Addgene | RRID:Addgene_17575 | GAL4 enhancer cloning vector for *Drosophila* |

*Continued on next page*

*Continued*

| Reagent type (species) or resource | Designation | Source or reference | Identifiers | Additional information |
|---|---|---|---|---|
| Sequence-based reagent | EO053-f | This paper | PCR primers | CCCGGAGCGGCACAATTAGTCTTG |
| Sequence-based reagent | EO053-r | This paper | PCR primers | CGGTAATGCTGAATGAACCTTTCAA |
| Sequence-based reagent | DvEO053-f | This paper | PCR primers | TGCCCTGGTTCTTTGGCTAACACG |
| Sequence-based reagent | DvEO053-r | This paper | PCR primers | TTTCTTGTACATAATCGTTCTTGG |
| Sequence-based reagent | Zen2US-f | This paper | PCR primers | TTATATACCCCAGAAGCCCTTCGTGACG |
| Sequence-based reagent | Zen2US-r | This paper | PCR primers | TGATGTGATGACACCAATTTATCTGAGC |
| Commercial assay or kit | LR Clonase II kit | Thermofisher | Cat# 11791020 | |
| Commercial assay or kit | TOPO pCR8/GW kit | Thermofisher | Cat# K2500-20 | |
| Commercial assay or kit | DIG RNA labelling mix | Roche | Cat#11277073910 | |
| Commercial assay or kit | Biotin RNA labelling mix | Roche | Cat#11685597910 | |
| Commercial assay or kit | T7 RNA polymerase | Roche | Cat. No. 10 881 767 001 | |
| Commercial assay or kit | MAXIscript T7 transcription kit | ThermoFisher | Cat# AM1312 | |
| Commercial assay or kit | mMESSAGE mMACHINE T7 Transcription kit | ThermoFisher | Cat# AM1344 | |
| Commercial assay or kit | SuperScript II Reverse Transcription Kit | ThermoFisher | Cat# 18064022 | |
| Commercial assay or kit | iQ SYBR Green Supermix | BioRad | Cat# 18064022 | |
| Other | NBT/BCIP stock solution | Roche | Cat#11681451001 | |

## Reporter constructs

The EO053 region was identified by enriched CBP binding in embryonic rather than later stages [i.e., 'Embryo Only' (EO) (*Nègre et al., 2011*)]. It was amplified from genomic DNA using the primers EO053-f (CCCGGAGCGGCACAATTAGTCTTG) and EO053-r (CGGTAATGCTGAATGAACCTTTCAA). ΔFG, FG, and TTAAm mutations were generated using overlap extension PCR (*Ho et al., 1989*). The DvEO053 primers were DvEO053-f (TGCCCTGGTTCTTTGGCTAACACG) and DvEO053-r (TTTCTTGTACATAATCGTTCTTGG). PCR products amplified from genomic DNA were cloned into pBPGUw (*Pfeiffer et al., 2008*) through an LR Clonase II (ThermoFisher—Waltham, MA) Gateway reaction from a pCR8/TOPO/GW intermediate. All variants were inserted upstream of the DSCP promoter in the same orientation with respect to EO053. zen2US was amplified using zen2US-f (TTATATACCCCAGAAGCCCTTCGTGACG) and zen2US-r (TGATGTGATGACACCAATTTATCTGAGC) and cloned as above. See also *Supplementary file 4* for complete sequences.

## Generation of *pb* mutants by CRISPR/Cas9 mutagenesis

Preparation of guide RNA and Cas9 mRNA was done as described previously (*Hwang et al., 2013*). The sequences for guide RNAs directed against EO053 (GGAGTCGGTCGGACACAGAG) or region 2.1 (GAGAAAGATTTTCTCCCCTC and GCTGTGCCTCATTTAATGCA) were cloned into DR274 (Addgene—Cambridge, MA; deposited by J Keith Joung) cut with BsaI. For $pb^{M2:20}$, sequence-verified clones were linearized with DraI and 1 µg transcribed using the MAXIscript T7 kit

(ThermoFisher). Cas9 mRNA was transcribed using the mMESSAGE mMachine T7 kit, using 1 µg MLM3613 (Addgene; deposited by J Keith Joung) linearized with PmeI. RNAs were precipitated according to the manufacturer's instructions and resuspended in injection buffer. The final injection cocktail for injecting $w^{1118}$ embryos contained 900 ng/µL Cas9 mRNA and 100 ng/µL EO053 guide RNA. For $pb^{11A}$, $pb^{11C}$, $pb^{11D}$, and $pb^{11E}$, $pb^{M2:20}$ flies were crossed to Cas9-expressing flies and a cocktail containing both guide RNAs and the homology-directed repair template were injected by BestGene, Inc The region 2.1 HDR template was constructed by separately cloning each arm (~2 kb each, see *Supplementary file 4*) into pGEM-T (Promega). The left arm was flanked by BamHI and SalI sites and the right arm contained tandem BglII and XhoI sites on the 5' end. The left arm was cut with BamHI and SalI and subcloned into the plasmid containing the right arm, digested with BglII and XhoI. Injected flies were crossed to *w;;TM2/TM6C* individually, and then F1 males from each injected fly were crossed individually to *w;;TM2/TM6C*. After viable larvae were detected, the F1 males were removed from the vials and pooled into groups of four for gDNA extraction and PCR screening for deletion. F2 vials from positive pools were screened and sequenced to uncover the $pb^{M2:20}$ 1255-bp deletion and the precise region 2.1 deletion. Only a single injected fly harbored the 2.1 deletion, and we were ecstatic to obtain progeny with deletions on both the EO053 deletion chromosome and the wildtype homologous chromosome to provide us with both the single and double mutants.

### In situ hybridization

The *GAL4* digoxygenin probe was prepared as previously described (*Nègre et al., 2011*; *Pfeiffer et al., 2008*). The large exons from *pb* and *zen2* were amplified from genomic DNA and cloned into pGEM-T (Promega—Madison, WI). Linearized plasmids served as template for in vitro transcription of digoxygenin-labeled RNA probes as described (*Kosman et al., 2004*) using T7 RNA polymerase (Promega or Roche) and DIG-UTP or biotin-UTP RNA labeling mixes (Roche). Embryo in situ hybridizations using *GAL4*, *LacZ*, *pb*, or *zen2* digoxygenin probes and *HLHmβ* biotin probes were performed as previously described (*Kosman et al., 2004*; *Nègre et al., 2011*; *Reeves and Posakony, 2005*). *GAL4* in situ hybridizations with related mutant constructs were performed in parallel batches and representative images presented.

### *pb* in situ hybridization image scoring

Following in situ hybridization and mounting of $w^{1118}$ and $pb^{M2:20}$ embryos in parallel, images were collected of lateral views of stage 10–12 embryos. Filenames of experimental and control in situ hybridization images were gathered, randomly shuffled, and renamed for double-blind scoring image analysis using ImageJ (Fiji). The area of visible stain in maxillary and labial segments was selected for each image and values for area, mean intensity, min intensity, and max intensity were recorded. After data collection, values were reassigned to the corresponding genotype and statistically analyzed.

### *zen2* in situ hybridization image scoring

Following in situ hybridization and mounting of $w^{1118}$ and $pb^{M2:20}$ embryos in parallel, slides were manually screened for dorsal visibility at stage 4 or stage 5 (staging based upon cellularization under DIC optics). The in situ hybridization signal at these stages was scored as 'dorsal-weak,' 'dorsal-strong,' 'poles+dorsal,' 'poles-strong,' 'poles-weak,' or 'no expression.' In *Figure 5*, 'dorsal-weak' and 'dorsal-strong' represent the 'dorsal' category, and 'poles+dorsal,' 'poles-strong,' and 'poles-weak' represent the 'polar' category, to distinguish between EO053-like and zen2US-like expression.

### Quantitative PCR

RNA was prepared using Trizol (Ambion) from embryos collected for 2 hr and aged 4 hr at 25 °C (4–6 hr embryos). Sample embryos were examined with a compound microscope to verify desired age (approximately stage 11). First-strand cDNA was synthesized with a SuperScript II kit (Invitrogen). Quantitative RT-PCR was performed on an iQ5 cycler (BioRad) using iQ SYBR Green Supermix (BioRad).

## Scaffold analysis

We queried genomic scaffolds of sequenced arthropods by BLAST using *Drosophila melanogaster* amino acid sequence for Pb, Zen2, Zen, or Dfd, as well as orthologous sequences from other annotated species. Scaffolds or accession numbers were obtained from http://metazoa.ensembl.org/, http://hymenopteragenome.org/, http://www.vectorbase.org/, http://i5k.nal.usda.gov/, http://www.ncbi.nlm.nih.gov/genbank/, http://genome.wustl.edu/, http://www.collembolomics.nl/collembolomics/. Gene structures were inferred from either annotations or database gene predictions. For some species, only the homeodomain sequence for each gene was determined and located. In some cases (e.g., *Dendroctonus ponderosae*), the *pb* and *zen* homeodomains are encoded on separate scaffolds but the *zen* scaffold includes near one end a YPWM-encoding ORF that by BLAST is most similar to *pb*, suggesting that the scaffolds are likely adjacent.

## Gene diagrams and motif analysis

Each sequence was opened in GenePalette (*Rebeiz and Posakony, 2004*; *Smith et al., 2017*), and the location of each gene identified by either GenBank or manual annotation. Sequences were searched for instances of ATCATTAATCAT (allowing for 1-bp mismatch), 'ATTAAT' (ATCATTAAT or ATTAATCAT), and 'YPWM' (TAYCCNTGGATG). Instances found at similar relative locations in related species were analyzed for similarity in both core and flanking sequence to suggest orthology within clades. All figure gene diagrams were generated by creating a postscript export from GenePalette and compiling/editing in Adobe Illustrator.

# Acknowledgements

We wish to acknowledge those members of the lab contributing helpful comments throughout the course of this work, including Jenny Atanasov and Artem Movsesyan. We are also grateful for the services of Genetic Services, Inc (Cambridge, MA), and BestGene, Inc (Chino Hills, CA) for injecting some of the constructs used in this study.

# Additional information

### Funding

| Funder | Grant reference number | Author |
|--------|------------------------|--------|
| NIH | 1R01GM120377 | James W Posakony |

The funders had no role in study design, data collection and interpretation, or the decision to submit the work for publication.

### Author contributions

Steve W Miller, Conceptualization; Investigation; Resources; Writing - original draft, review, and editing; Visualization; James W Posakony, Conceptualization; Supervision; Funding acquisition; Writing - original draft, review, and editing

### Author ORCIDs

Steve W Miller (iD) https://orcid.org/0000-0001-7610-6336
James W Posakony (iD) https://orcid.org/0000-0001-6377-1732

### Decision letter and Author response

Decision letter https://doi.org/10.7554/eLife.39876.sa1
Author response https://doi.org/10.7554/eLife.39876.sa2

# Additional files

## Supplementary files

• Supplementary file 1. Occurrence and conservation of Hox-like binding motifs in the *pb* region across Arthropods. Phylogenetic compilation of motifs similar to the conserved EO053 Hox-like binding motif in the *pb* region. Alignments correspond to each of the numbered motifs in *Figure 5— figure supplement 3*–9; shown is a 24-nt segment including each motif (core 12 nt flanked by 6 nt on each side).

• Supplementary file 2. Instances of random 12-mer occurrences in the *pb* region across Schizophora. A list of 40 random 12-mers analyzed for instances of occurrence and conservation between different species. We also picked a random set of 10 12-mers based upon conservation within the large *pb* intron between *D. melanogaster* and *D. ananassae* for patterns of extended conservation in Schizophora. Only one of the *ananassae* 12-mers shows conservation beyond *D. grimshawi*.

• Supplementary file 3. Promoter sequences relevant to *Figure 7—figure supplement 1*. Shown are the *D. melanogaster* and *D. virilis* endogenous promoters for *pb* and *zen2*, noting the presence and location of various promoter elements, as well as the DSCP sequence from the pBPGUw reporter vector (*Pfeiffer et al., 2008*). Secondly are shown the sequences replacing the DSCP promoter for testing promoter influence on enhancer expression: *pb*, *zen2*-modified DSCP, and *bcd*.

• Supplementary file 4. Reporter sequences and $pb^{M2:20}$ deletion breakpoints. All PCR-amplified sequences shown were cloned into pCR8/GW/TOPO. Sequence-verified clones with the same orientation as EO053wt were integrated into pBPGUw by Gateway reaction using LR Clonase II (*Pfeiffer et al., 2008*). Also included are the sequences at the end of each breakpoint of the $pb^{M2:20}$ deletion and the HDR template sequence used to generate the deletion of region 2.1.

• Transparent reporting form

## Data availability

No new sequence data generated, all found in public archives (GenBank, Ensembl, and other public genome browsers/queries). Accession numbers for all genome scaffolds used are collected in Fig. 6 - Figure Supplements 1-9 - Source Data.

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
