## [Decision Letter]

[Editorial note: This article has been through an editorial process in which the authors decide how to respond to the issues raised during peer review. The Reviewing Editor's assessment is that all the issues have been addressed.]

Thank you for submitting your article "Disparate expression specificities coded by a shared Hox-C enhancer" for consideration by *eLife*. Your article has been reviewed by three peer reviewers, and the evaluation has been overseen by Patricia Wittkopp as the Senior and Reviewing Editor. The following individuals involved in review of your submission have agreed to reveal their identity: Justin Crocker (Reviewer #1); Michael W Perry (Reviewer #3).

The Reviewing Editor has highlighted the concerns that require revision and/or responses, and we have included the separate reviews below for your consideration. If you have any questions, please do not hesitate to contact us.

Summary:

In this paper, Miller and Posakony show an interesting enhancer case study that differs from the two primary models of enhancer functions: an enhancer that controls a single gene and drives a specific pattern or a locus control region that controls multiple genes with similar patterns. Their case study is of an enhancer that controls two genes with different outputs. The authors create a large set of transgenic lines to show these two activities are not easily separated into different regions of the enhancer, generate an enhancer deletion to show the region affects both genes, and perform a comparative sequence analysis to look for evidence of evolutionary conservation. Overall, the paper describes this case study well and is an important contribution to the enhancer biology field.

Interestingly, this case is almost the opposite of what has been previously proposed, where "seemingly redundant" or "shadow" enhancers drive the same pattern of expression and provide additional flexibility. In that case, modification of either enhancer might not disrupt the core ancestral function. In this case, having a single enhancer shared between genes might instead constrain the system. The existence of this kind of regulatory arrangement has striking implications for the evolution of gene regulation.

Essential revisions:

1) Alternative promoters: Please show the expression data from the experiment referred to in the text in which the synthetic core promoter was replaced with the endogenous promoters. Be sure to describe the coordinates of the promoter region used.

2) Quantification: All reviewers and the editor identified the lack of quantification of expression patterns as a weakness of this study. At the same time, the majority view of the reviewers was that it wasn't essential to repeat all the expression analyses to make them quantitative. The very different patterns of the two genes are clearly different without quantification. In cases where a distinct portion of the pattern is always present or absent (as in much of the deletion analysis), colormetric in situs are probably fine. Not a lot of specific conclusions are taken from most of the FG# lines in any case, the pattern is simply reported. Things become slightly more problematic when the pattern is variable across embryos at the same stage, or when they assign a score and count embryos, as in Figure 5. Figure 5 is the centerpiece of the manuscript, but it falls short of meeting the standards of the field. For example, it is not clear what the binning of the *pb* in situs represents. Classical colorimetric stains are not always linear reactions so that the data could be compressed, threshold, etc. The authors should repeat with controlled conditions and fluorescent antibodies. Ideally, they would also do further analysis to look at variability, noise, etc.

As one reviewer wrote: the use of in situs to determine the effect of the *pb^M2:20^*deletion isn't the most accurate approach to measuring mRNA in the embryo. This result would be greatly strengthened by a complementary, more quantitative approach, e.g. qPCR, single molecule FISH, or an in situ with a normalization gene stained for reference. Alternatively, another method to show the enhancer loops to both the *pb* and *zen2* promoters might be used. Statements such as "weakly," rare, etc. make it hard to evaluate the authors' findings.

3) Phenotypes: What are the phenotypic consequences of the CRISPR deletion of the endogenous enhancer? Are the embryos always okay? Are there fitness consequences? What about robustness? The authors could even do classical cuticle preps to look at patterning defects. Each of these is very compelling questions, and they have a beautiful platform to study these questions. It is a shame for the authors to drop the ball on this!

4) Please provide the precise sequences of the transgenic constructs (in particular the truncations and mutations). They will be useful for future sequence analysis.

Although the separate reviews are presented below in their entirety, I have also prepared this consolidated set of additional suggestions. Although redundant with the original reviews, I thought it might still be helpful to have the requests synthesized all in one place:

Additional comments:

1) Starting with Figure 1 and related text; double in situs would be ideal, and they should at least put images from a public repository to aid the readers who would not want to chase down the original publications. Minor note, in the related text the authors call them blastoderm embryos while the figure is not.

2) Relevant other literature to cite: (1) There is an example of a related phenomena in chick, where an enhancer drives two different spatiotemporal patterns of expression, but control a single gene. It may be useful to reference this: https://www.ncbi.nlm.nih.gov/pubmed/21775416. (2) This story reminds me of work from the Kassis lab (Cheng et al., 2014) on the gene engrailed and invected. They also used deletions at the locus to show that single regulatory regions may regulate both genes. This system is if anything even more complex and lacks the evolutionary components of the current manuscript, but because of the similarity should be mentioned and cited. (3) There are examples of a single enhancer engaging two promoters simultaneously (https://www.ncbi.nlm.nih.gov/pubmed/27293191), so the temporal separation of activities may not be strictly needed. (4) Possible citation to consider; Cande, Goltsev, and Levine PNAS 2009 also discuss microsynteny resulting from enhancer position in the intron of a neighboring gene.

3) In the section discussing the motif conservation, it would strengthen the result to produce more than one 12mer as a negative control.

4) In paragraph two of subsection “Distinct temporal and spatial specificities”, the authors should consider the possibility of missing dominant repressors. These could be located either near the promoter but outside the region tested with their reporter (as mentioned) or even at another enhancer. Dominant repression appears to be a not uncommon feature; exe. the gap genes each seem to be influenced by the hkb terminal repressor which acts in a dominant fashion. These binding sites are sometimes located at alternate or shadow enhancers; see Perry, Boettiger and Levine, 2011.

5) Figure 2 and 3: It could aid readers to call the truncations "minimal enhancers" or "elements" and then the deletion "delFG."

6) Regarding the Discussion, the authors could also consider that these genes are in a shared neighborhood of expression. It has been speculated that the average enhancer has a range of activities in which it can influence any active promoter within its reach (see, for example, Quintero-Cadena and Sternberg, 2016). Therefore, this could be noise in the system that is tolerated by the embryo – going back to my questions about phenotypic differences.

7) In the fourth paragraph of the Introduction, the sentence starting "Intriguingly…" might be re-worded for clarity. At first read, it's hard to understand how the enhancer drives a pattern that resembles two genes with different patterns. Perhaps indicate that the enhancer drives a pattern that represents a combination or a union of the two genes' expression patterns?

8) In Figure 1, can you make the heights of the two enhancer regions equivalent? I don't believe that the different heights are meaningful.

9) In Figure 2 and 3, I believe the red region corresponding to part of trunc 3 indicates the repression of late DV/AS expression, but this isn't clear from the diagram or legends. Also, could you indicate on the figure itself the meaning of the asterisk and cross?

10) I couldn't see the *pb*-like expression in panel 2J very well on a printed version of the figure. Is there a way to make this clearer? A zoomed in inset?

11) Can you add a key to Figure 4A to indicate which mutations affect the *pb* pattern and which affect the *zen* pattern (or both)?

12) Abstract fourth sentence: could delete "as well"

13) The embryos in Figure 1 are cropped well, Figure 2 and 3 are not. Figure 4 is a little better. It's hard enough to see small patterns, please help the reader and crop as much as possible. Perhaps consider showing only the anterior half of the embryo in cases like the right two columns of Figure 4.

14) Figure 2 legend: ventral, not vessel

15) Figure 7: just a suggestion to consider – it's easier to interpret diagrams of genes running from left to right, regardless of their orientation in the genome. The gene model is already a little smashed; it might be easier to tell what was going on if it were shown left to right, promoter arrow up top. It might also help to include and label each enhancer whether or not it is active and to rely on the looping or arrows of interaction to show activity. The PREs should either be left out (not necessary or relevant for this model) or at least made to look different than the two enhancers; otherwise they look like unlabeled enhancers. They were shown clearly enough in an earlier figure and should probably be removed here.

The following two points were not considered essential revisions, but would strengthen the paper:

1) The authors only really take things down to the binding site level for one particularly suggestive but somewhat generic motif. It would be a wonderful addition if they were able to either a) identify the factor(s) that binds this evolutionarily conserved site using genetic approaches (perhaps by examining reporter expression in candidate mutant backgrounds) or b) test the effect of mutating this site directly.

2) The single additional experiment I would most like to see to strengthen their assertion that a conserved site influences the real expression of both genes would be to evaluate the effects of a CRISPR modification of just this binding site, either scrambling or removing it. It remains formally possible that the EO053 region contains two intercalated enhancers that do not share physical binding sites. Perhaps the second evolved in the same position simple because it is accessible. Any additional binding sites could have been scrambled beyond recognition via conservation analysis by compensatory evolution even within the *Drosophila* genus (as in several papers on the eve locus). That leaves this one site that is so deeply conserved, but the manuscript lacks a direct test of its function.

3) In the evolutionary section the authors describe deep conservation but also many mismatches to the conserved motif they follow. It would be nice to know whether these differences are ever functional, but this is probably beyond the scope of this paper. Matching endogenous patterns to reporters in those same species (to avoid trans effects) quickly becomes a difficult prospect.

Separate reviews (please respond to each point):

Reviewer #1:

In the manuscript "Disparate expression specificities coded by a shared Hox-C enhancer" Miller and Posakony explore how a single regulatory sequence is shared by two genes that undergo functional divergence. Specifically, they find that they are unable to separate the *pb*-like and *zen2*-like specificities within a share regulatory region. Furthermore, deletion of the shared enhancer affects the expression of both genes. Taken together, a nice demonstration of two genes that have evolved different outputs while sharing an enhancer.

I have several experimental and editorial changes that would enhance this manuscript.

Starting with Figure 1 and related text; double in situs be ideal, and they should at least put images from a public repository to aid the readers who would not want to chase down the original publications. Minor note, in the related text the authors call them blastoderm embryos while the figure is not.

Figure 2: It could aid readers to call the truncations "minimal enhancers" or "elements." While it may be difficult at this point, it would have been nice to have some quantification. Statements such as "weakly," rare, etc. make it hard to evaluate the authors' findings.

Figure 3: again, I recommend calling the truncations "elements" and then the deletion "delFG."

Figure 5: This is the centerpiece of the manuscript, but it falls short of meeting the standards of the field. For example, it is not clear what the binning of the *pb* in situs represents. Classical colorimetric stains are not always linear reactions so that the data could be compressed, threshold, etc. The authors should repeat with controlled conditions and fluorescent antibodies. Ideally, they would also do further analysis to look at variability, noise, etc.

Finally, they did not talk at all about any phenotypic consequences. Are the embryos always okay? Are there fitness consequences? What about robustness. The authors could even do classical cuticle preps to look at patterning defects. Each of these is very compelling questions, and they have a beautiful platform to study these questions. It is a shame for the authors to drop the ball on this!

Regarding the Discussion, the authors could also consider that these genes are in a shared neighborhood of expression. It has been speculated that the average enhancer has a range of activities in which it can influence any active promoter within its reach (see, for example, Quintero-Cadena and Sternberg, 2016). Therefore, this could be noise in the system that is tolerated by the embryo – going back to my questions about phenotypic differences.

In sum, it is a compelling series of experiments. I wish the authors would have taken the extra effort to finalize the experiments to nail down their original question regarding shared enhancers. I would even host any interested in my group, providing reagents to see the results!

Reviewer #2:

In this paper, Miller and Posakony show an interesting enhancer case study that differs from the two primary models of enhancer functions: an enhancer that controls a single gene and drives a specific pattern or a locus control region that controls multiple genes with similar patterns. Their case study is of an enhancer that controls two genes with different outputs. The authors create a large set of transgenic lines to show these two activities are not easily separated into different regions of the enhancer, generate an enhancer deletion to show the region affects both genes (see point #2 below), and perform a comparative sequence analysis to look for evidence of evolutionary conservation. Overall, the paper describes this case study well and is an important contribution to the enhancer biology field. I have a few suggestions to strengthen the claims of the paper:

Suggestions:

1) There is an example of a related phenomena in chick, where an enhancer drives two different spatiotemporal patterns of expression, but control a single gene. It may be useful to reference this: https://www.ncbi.nlm.nih.gov/pubmed/21775416.

2) The use of in situs to determine the effect of the *pb^M2:20^*deletion isn't the most accurate approach to measuring mRNA in the embryo. This result would be greatly strengthened by a complementary, more quantitative approach, e.g. qPCR, single molecule FISH, or an in situ with a normalization gene stained for reference. Alternatively, another method to show the enhancer loops to both the *pb* and *zen2* promoters might be used.

3) In the section discussing the motif conservation, it would strengthen the result to produce more than one 12mer as a negative control.

4) There are examples of a single enhancer engaging two promoters simultaneously (https://www.ncbi.nlm.nih.gov/pubmed/27293191), so the temporal separation of activities may not be strictly needed.

5) It would useful to see the expression data when the synthetic core promoter is replaced with the endogenous promoters. How large of a promoter region was used?

6) I couldn't find the precise sequences of the transgenic constructs (in particular the truncations and mutations). It would be useful to provide these to allow future sequence analysis.

Minor Comments:

– In the fourth paragraph of the Introduction, the sentence starting "Intriguingly…" might be re-worded for clarity. At first read, it's hard to understand how the enhancer drives a pattern that resembles two genes with different patterns. Perhaps indicate that the enhancer drives a pattern that represents a combination or a union of the two genes' expression patterns?

– In Figure 1, can you make the heights of the two enhancer regions equivalent? I don't believe that the different heights are meaningful.

– In Figure 2 and 3, I believe the red region corresponding to part of trunc 3 indicates the repression of late DV/AS expression, but this isn't clear from the diagram or legends. Also, could you indicate on the figure itself the meaning of the asterisk and cross?

– I couldn't see the *pb*-like expression in panel 2J very well on a printed version of the figure. Is there a way to make this clearer? A zoomed in inset?

– Can you add a key to Figure 4A to indicate which mutations affect the *pb* pattern and which affect the *zen* pattern (or both)?

Reviewer #3:

In this manuscript Miller and Posakony characterize a new *Drosophila* embryonic enhancer in the Hox complex and provide evidence that it is novel in an interesting way: it is a single enhancer shared by two genes. It drives expression in the distinctive patterns of two neighboring genes and they use a classic deletion analysis to uncover subregions that influence one or both patterns. Next, they show that CRISPR deletion of the endogenous enhancer influences expression of both neighboring genes, despite the existence of other enhancers that are known to drive overlapping expression patterns. Finally, they use sequence conservation to argue that this shared regulatory relationship may extend over relatively deep evolutionary timescales.

This is a new and interesting variation on the theme that the number of enhancers that produce a particular pattern might influence how novel or modified patterns might evolve. In this case it's almost the opposite what has been previously proposed, where "seemingly redundant" or "shadow" enhancers drive the same pattern of expression and provide additional flexibility. In that case, modification of either enhancer might not disrupt the core ancestral function. In this case, having a single enhancer shared between genes might instead constrain the system. The existence of this kind of regulatory arrangement has striking implications for the evolution of gene regulation.

I find the work to be clearly written and of broad general interest. I have several minor comments and concerns but in general would be happy to support publication.

First, the authors only really take things down to the binding site level for one particularly suggestive but somewhat generic motif. It would be a wonderful addition if they were able to either a) identify the factor(s) that binds this evolutionarily conserved site using genetic approaches (perhaps by examining reporter expression in candidate mutant backgrounds) or b) test the effect of mutating this site directly. The single additional experiment I would most like to see to strengthen their assertion that a conserved site influences the real expression of both genes would be to evaluate the effects of a CRISPR modification of just this binding site, either scrambling or removing it. It remains formally possible that the EO053 region contains two intercalated enhancers that do not share physical binding sites. Perhaps the second evolved in the same position simple because it is accessible. Any additional binding sites could have been scrambled beyond recognition via conservation analysis by compensatory evolution even within the *Drosophila* genus (as in several papers on the eve locus). That leaves this one site that is so deeply conserved, but the manuscript lacks a direct test of its function.

Next, this story reminds me of work from the Kassis lab (Cheng et al., 2014) on the gene engrailed and invected. They also used deletions at the locus to show that single regulatory regions may regulate both genes. This system is if anything even more complex and lacks the evolutionary components of the current manuscript, but because of the similarity should be mentioned and cited.

In the evolutionary section the authors describe deep conservation but also many mismatches to the conserved motif they follow. It would be nice to know whether these differences are ever functional, but this is probably beyond the scope of this paper. Matching endogenous patterns to reporters in those same species (to avoid trans effects) quickly becomes a difficult prospect.

In paragraph two of subsection “Distinct temporal and spatial specificities”, the authors should consider the possibility of missing dominant repressors. These could be located either near the promoter but outside the region tested with their reporter (as mentioned) or even at another enhancer. Dominant repression appears to be a not uncommon feature; exe. the gap genes each seem to be influenced by the hkb terminal repressor which acts in a dominant fashion. These binding sites are sometimes located at alternate or shadow enhancers; see Perry, Boettiger and Levine, 2011.

Minor Comments:

Abstract fourth sentence: could delete "as well"

The embryos in Figure 1 are cropped well, Figure 2 and 3 are not. Figure 4 is a little better. It's hard enough to see small patterns, please help the reader and crop as much as possible. Perhaps consider showing only the anterior half of the embryo in cases like the right two columns of Figure 4.

Figure 2 legend: ventral, not vessel

Possible citation to consider; Cande, Goltsev, and Levine PNAS 2009 also discuss microsynteny resulting from enhancer position in the intron of a neighboring gene.

Figure 7: just a suggestion to consider – it's easier to interpret diagrams of genes running from left to right, regardless of their orientation in the genome. The gene model is already a little smashed; it might be easier to tell what was going on if it were shown left to right, promoter arrow up top. It might also help to include and label each enhancer whether or not it is active and to rely on the looping or arrows of interaction to show activity. The PREs should either be left out (not necessary or relevant for this model) or at least made to look different than the two enhancers; otherwise they look like unlabeled enhancers. They were shown clearly enough in an earlier figure and should probably be removed here.

---

## [Author Response]

Essential revisions:1) Alternative promoters: Please show the expression data from the experiment referred to in the text in which the synthetic core promoter was replaced with the endogenous promoters. Be sure to describe the coordinates of the promoter region used.

We have included an annotated Supplementary file with the sequences of the promoter alterations, along with images of the reporter expression.

2) Quantification: All reviewers and the editor identified the lack of quantification of expression patterns as a weakness of this study. At the same time, the majority view of the reviewers was that it wasn't essential to repeat all the expression analyses to make them quantitative. The very different patterns of the two genes are clearly different without quantification. In cases where a distinct portion of the pattern is always present or absent (as in much of the deletion analysis), colormetric in situs are probably fine. Not a lot of specific conclusions are taken from most of the FG# lines in any case, the pattern is simply reported. Things become slightly more problematic when the pattern is variable across embryos at the same stage, or when they assign a score and count embryos, as in Figure 5. Figure 5 is the centerpiece of the manuscript, but it falls short of meeting the standards of the field. For example, it is not clear what the binning of the pb in situs represents. Classical colorimetric stains are not always linear reactions so that the data could be compressed, threshold, etc. The authors should repeat with controlled conditions and fluorescent antibodies. Ideally, they would also do further analysis to look at variability, noise, etc.As one reviewer wrote: the use of in situs to determine the effect of the pb^M2:20^ deletion isn't the most accurate approach to measuring mRNA in the embryo. This result would be greatly strengthened by a complementary, more quantitative approach, e.g. qPCR, single molecule FISH, or an in situ with a normalization gene stained for reference. Alternatively, another method to show the enhancer loops to both the pb and zen2 promoters might be used. Statements such as "weakly," rare, etc. make it hard to evaluate the authors' findings.

We thank the reviewers for the critical analysis of this set of experiments. We felt that the scoring of embryos based upon stain demonstrated the variability that was being asked for but perhaps the question was more directed at the use of colorimetric staining. It is true that colorimetric staining is subject to saturation with extended duration, but that would skew our results toward occluding differences rather than providing false-positive results. For each experiment, all in situs were performed in parallel to account for variations in embryo exposure and probe accessibility. In addition, the examination of a large number of embryos between mutant and wild-type would also incorporate parallel variables within the data. To present an alternative analysis of these images that removes the subjective binning, we took the same images and measured the pixel intensity within the area stained on the maxillary and labial segments in each embryo (lower intensity reflects a darker stain) and are reporting the average intensity of the area measured per embryo as well as the average minimum intensity per embryo of each genotype. As with our previous analysis, each picture was analyzed blind to the identity of its genotype.

Given the proximity of the *pb* and *zen2* promoters there is a strong chance of getting a false-positive result with a 3C-like approach. This is also why we felt compelled to invest in expression analysis. qPCR will only detect a reliable difference in expression levels of at least 1.5-fold or greater. Nevertheless, we report qPCR analysis and indeed fail to detect differences between *w1118* and *pb^M2:20^*.

We suspected that the other enhancer, region 2.1, mentioned in our analysis was likely a key regulatory region and was potentially occluding any effect of deleting EO053 alone. For this reason, we chose to delete this region as well. We not only see that deleting region 2.1 alone is sufficient to observe a consistent visible difference in expression of *pb*, but also that deleting EO053 produces an even stronger phenotype. We feel confident that these additional experiments will alleviate concerns by the reviewers about the strength of our data on the role of EO053 on *pb* expression.

3) Phenotypes: What are the phenotypic consequences of the CRISPR deletion of the endogenous enhancer? Are the embryos always okay? Are there fitness consequences? What about robustness? The authors could even do classical cuticle preps to look at patterning defects. Each of these is very compelling questions, and they have a beautiful platform to study these questions. It is a shame for the authors to drop the ball on this!

We certainly understand the question, but we would be surprised to observe any phenotype at all, actually. It has long been known that *pb* is unique among *Drosophila* Hox genes in that it is lacking an embryonic patterning defect. Specifically, embryos carrying deletions spanning both the *zen2* and *pb* loci fail to show cuticle defects (https://www.ncbi.nlm.nih.gov/pubmed/2850265). We have added text to this effect to the section related to Figure 5.

Moreover, as we mentioned in the text, *pb* mini-genes deleting most of the intron (including EO053) successfully rescue adult mouthpart transformations (https://www.ncbi.nlm.nih.gov/pubmed/7635058). In this analysis it was only upon the deletion of both enhancers (deletion of the 2.1 fragment within the context of the mini-gene rescue construct) that a phenotype was observed. We suspected the 2.1 fragment is likely making a stronger contribution to *pb* expression, and the newly added findings from the additional deletion experiments we have subsequently performed demonstrate the strong phenotypic consequence of loss of region 2.1 only.

4) Please provide the precise sequences of the transgenic constructs (in particular the truncations and mutations). They will be useful for future sequence analysis.

We have added a Supplementary file with the sequences of the constructs, as well as the breakpoint-adjacent sequences for *pb^M2:20^* (Supplementary file 4).

Although the separate reviews are presented below in their entirety, I have also prepared this consolidated set of additional suggestions. Although redundant with the original reviews, I thought it might still be helpful to have the requests synthesized all in one place:Additional comments:1) Starting with Figure 1 and related text; double in situs would be ideal, and they should at least put images from a public repository to aid the readers who would not want to chase down the original publications. Minor note, in the related text the authors call them blastoderm embryos while the figure is not.

Thank you for the suggestion. Since we later show in situ expression patterns for both *pb* and *zen2* in Figure 5 we included this reference in the Figure 1 legend. We also added links to the independent BDGP in situ pages for each of these genes to Figure 1 legend.

As to the second point, we have made changes to the text to make use of the term “blastoderm” less loosely when referencing the *zen2*-like expression.

2) Relevant other literature to cite: (1) There is an example of a related phenomena in chick, where an enhancer drives two different spatiotemporal patterns of expression, but control a single gene. It may be useful to reference this: https://www.ncbi.nlm.nih.gov/pubmed/21775416. (2) This story reminds me of work from the Kassis lab (Cheng et al., 2014) on the gene engrailed and invected. They also used deletions at the locus to show that single regulatory regions may regulate both genes. This system is if anything even more complex and lacks the evolutionary components of the current manuscript, but because of the similarity should be mentioned and cited. (3) There are examples of a single enhancer engaging two promoters simultaneously (https://www.ncbi.nlm.nih.gov/pubmed/27293191), so the temporal separation of activities may not be strictly needed. (4) Possible citation to consider; Cande, Goltsev, and Levine PNAS 2009 also discuss microsynteny resulting from enhancer position in the intron of a neighboring gene.

Thank you for these suggestions to improve the manuscript; we have made appropriate inclusions.

3) In the section discussing the motif conservation, it would strengthen the result to produce more than one 12mer as a negative control.

We examined 40 random 12mers, and of these only a single site exhibited conservation out to virilis. We also examined 10 random intronic 12-mer sequences outside of EO053 that are conserved between *melanogaster* and *ananassae*, and found that only one exhibited conservation beyond *grimshawii*.

4) In paragraph two of subsection “Distinct temporal and spatial specificities”, the authors should consider the possibility of missing dominant repressors. These could be located either near the promoter but outside the region tested with their reporter (as mentioned) or even at another enhancer. Dominant repression appears to be a not uncommon feature; exe. the gap genes each seem to be influenced by the hkb terminal repressor which acts in a dominant fashion. These binding sites are sometimes located at alternate or shadow enhancers; see Perry, Boettiger and Levine, 2011.

We thank the reviewer for the suggestion, and have added this reference to this section.

5) Figure 2 and 3: It could aid readers to call the truncations "minimal enhancers" or "elements" and then the deletion "delFG."

We thank the reviewer for the suggestion, but wanted to make clear that all of these constructs represent subsets of the full-length enhancer, and thus are all “truncated” versions.

6) Regarding the Discussion, the authors could also consider that these genes are in a shared neighborhood of expression. It has been speculated that the average enhancer has a range of activities in which it can influence any active promoter within its reach (see, for example, Quintero-Cadena and Sternberg, 2016). Therefore, this could be noise in the system that is tolerated by the embryo – going back to my questions about phenotypic differences.

We agree that several lines of evidence support a model that within local regions there are often co-regulated genes, and the reference provided is indeed another in this grouping. We suggest it is highly likely that shared regulation between neighboring genes with overlapping expression patterns was the ancestral state of the Hox2-Hox3 region. Where we draw the contrast, however, is that these two genes have diverged in their expression, which would present an interesting challenge to this regulatory arrangement. A common assumption is that in response to such a challenge an enhancer would likely diverge to serve only one of the two genes, given the change in specificity. The key point of our study is to provide support to a model that under certain circumstances regulatory sharing can indeed be maintained despite change in specificity.

7) In the fourth paragraph of the Introduction, the sentence starting "Intriguingly…" might be re-worded for clarity. At first read, it's hard to understand how the enhancer drives a pattern that resembles two genes with different patterns. Perhaps indicate that the enhancer drives a pattern that represents a combination or a union of the two genes' expression patterns?

In response to this reviewer’s suggestion, we explored several changes to the text. We hope that any confusion in this sentence is alleviated by further investigation of the work, but we made a slight change to this sentence that we hope will suffice.

8) In Figure 1, can you make the heights of the two enhancer regions equivalent? I don't believe that the different heights are meaningful.

Indeed, the heights offer no point of significance. The only intention is to highlight the locations of the enhancers along the horizontal axis. We felt the area around the gene names to be somewhat crowded, and thus moved 2.1 up to improve discernment.

9) In Figure 2 and 3, I believe the red region corresponding to part of trunc 3 indicates the repression of late DV/AS expression, but this isn't clear from the diagram or legends. Also, could you indicate on the figure itself the meaning of the asterisk and cross?

We thank the reviewer for the suggested improvements. We have updated these figures to improve clarity.

10) I couldn't see the pb-like expression in panel 2J very well on a printed version of the figure. Is there a way to make this clearer? A zoomed in inset?

In response to this suggestion we have added insets to both 2I and 2J.

11) Can you add a key to Figure 4A to indicate which mutations affect the pb pattern and which affect the zen pattern (or both)?

In response to this suggestion we have made a modification of Figure 4 to include this, although next to the images, rather than in 4A. We have also chosen to show only the maxillary and labial segments rather than whole embryos for the ease of the reader.

12) Abstract fourth sentence: could delete "as well"

We deleted this text as suggested.

13) The embryos in Figure 1 are cropped well, Figure 2 and 3 are not. Figure 4 is a little better. It's hard enough to see small patterns, please help the reader and crop as much as possible. Perhaps consider showing only the anterior half of the embryo in cases like the right two columns of Figure 4.

We have adjusted figures 2 and 3 as suggested.

14) Figure 2 legend: ventral, not vessel

“Dorsal vessel” is correct, referring to the embryonic anatomical structure.

15) Figure 7: just a suggestion to consider – it's easier to interpret diagrams of genes running from left to right, regardless of their orientation in the genome. The gene model is already a little smashed; it might be easier to tell what was going on if it were shown left to right, promoter arrow up top. It might also help to include and label each enhancer whether or not it is active and to rely on the looping or arrows of interaction to show activity. The PREs should either be left out (not necessary or relevant for this model) or at least made to look different than the two enhancers; otherwise they look like unlabeled enhancers. They were shown clearly enough in an earlier figure and should probably be removed here.

We have modified this figure to indicate enhancer activity, and labeled the PREs to not confuse them with enhancers, as well as labeled the relevant enhancers in all stages to again distinguish from the PREs.

The following two points were not considered essential revisions, but would strengthen the paper:1) The authors only really take things down to the binding site level for one particularly suggestive but somewhat generic motif. It would be a wonderful addition if they were able to either a) identify the factor(s) that binds this evolutionarily conserved site using genetic approaches (perhaps by examining reporter expression in candidate mutant backgrounds) or b) test the effect of mutating this site directly.

We have included a mutant version of the enhancer mutating this site, and indeed show a strong effect of this mutation upon enhancer activity.

2) The single additional experiment I would most like to see to strengthen their assertion that a conserved site influences the real expression of both genes would be to evaluate the effects of a CRISPR modification of just this binding site, either scrambling or removing it. It remains formally possible that the EO053 region contains two intercalated enhancers that do not share physical binding sites. Perhaps the second evolved in the same position simple because it is accessible. Any additional binding sites could have been scrambled beyond recognition via conservation analysis by compensatory evolution even within the *Drosophila* genus (as in several papers on the eve locus). That leaves this one site that is so deeply conserved, but the manuscript lacks a direct test of its function.

We believe this point has now been addressed by included the mutation of the specific motif.

3) In the evolutionary section the authors describe deep conservation but also many mismatches to the conserved motif they follow. It would be nice to know whether these differences are ever functional, but this is probably beyond the scope of this paper. Matching endogenous patterns to reporters in those same species (to avoid trans effects) quickly becomes a difficult prospect.

We agree with the reviewer that this is probably beyond the scope of this paper. Without knowing the factor (although the strongest candidate is Deformed), we don’t know range of tolerated binding sites. We fully accept the possibility that mismatches included may not be bound by same factor, but an analysis of the degree of degeneracy in Dfd binding across Arthropods is best left for future investigations.

Separate reviews (please respond to each point):Reviewer #1:In the manuscript "Disparate expression specificities coded by a shared Hox-C enhancer" Miller and Posakony explore how a single regulatory sequence is shared by two genes that undergo functional divergence. Specifically, they find that they are unable to separate the pb-like and zen2-like specificities within a share regulatory region. Furthermore, deletion of the shared enhancer affects the expression of both genes. Taken together, a nice demonstration of two genes that have evolved different outputs while sharing an enhancer.I have several experimental and editorial changes that would enhance this manuscript. […] In sum, it is a compelling series of experiments. I wish the authors would have taken the extra effort to finalize the experiments to nail down their original question regarding shared enhancers. I would even host any interested in my group, providing reagents to see the results!

All addressed above.

Reviewer #2:[…] I have a few suggestions to strengthen the claims of the paper […]

All addressed above.

Reviewer #3:[…] First, the authors only really take things down to the binding site level for one particularly suggestive but somewhat generic motif. It would be a wonderful addition if they were able to either a) identify the factor(s) that binds this evolutionarily conserved site using genetic approaches (perhaps by examining reporter expression in candidate mutant backgrounds) or b) test the effect of mutating this site directly. The single additional experiment I would most like to see to strengthen their assertion that a conserved site influences the real expression of both genes would be to evaluate the effects of a CRISPR modification of just this binding site, either scrambling or removing it. It remains formally possible that the EO053 region contains two intercalated enhancers that do not share physical binding sites. Perhaps the second evolved in the same position simple because it is accessible. Any additional binding sites could have been scrambled beyond recognition via conservation analysis by compensatory evolution even within the *Drosophila* genus (as in several papers on the eve locus). That leaves this one site that is so deeply conserved, but the manuscript lacks a direct test of its function.

Addressed with the TTAAm mutant.

Next, this story reminds me of work from the Kassis lab (Cheng et al., 2014) on the gene engrailed and invected. They also used deletions at the locus to show that single regulatory regions may regulate both genes. This system is if anything even more complex and lacks the evolutionary components of the current manuscript, but because of the similarity should be mentioned and cited.

Addressed above.

In the evolutionary section the authors describe deep conservation but also many mismatches to the conserved motif they follow. It would be nice to know whether these differences are ever functional, but this is probably beyond the scope of this paper. Matching endogenous patterns to reporters in those same species (to avoid trans effects) quickly becomes a difficult prospect.

We agree with the reviewer that such analysis is beyond the scope of the paper.

In paragraph two of subsection “Distinct temporal and spatial specificities”, the authors should consider the possibility of missing dominant repressors. These could be located either near the promoter but outside the region tested with their reporter (as mentioned) or even at another enhancer. Dominant repression appears to be a not uncommon feature; exe. the gap genes each seem to be influenced by the hkb terminal repressor which acts in a dominant fashion. These binding sites are sometimes located at alternate or shadow enhancers; see Perry, Boettiger and Levine, 2011.

Addressed above.

Minor Comments:Abstract fourth sentence: could delete "as well"The embryos in Figure 1 are cropped well, Figure 2 and 3 are not. Figure 4 is a little better. It's hard enough to see small patterns, please help the reader and crop as much as possible. Perhaps consider showing only the anterior half of the embryo in cases like the right two columns of Figure 4.

We have adjusted the cropping of Figure 2 and 3 with the reviewer’s suggestion.

Figure 2 legend: ventral, not vesselPossible citation to consider; Cande, Goltsev, and Levine PNAS 2009 also discuss microsynteny resulting from enhancer position in the intron of a neighboring gene.Figure 7: just a suggestion to consider – it's easier to interpret diagrams of genes running from left to right, regardless of their orientation in the genome. The gene model is already a little smashed; it might be easier to tell what was going on if it were shown left to right, promoter arrow up top. It might also help to include and label each enhancer whether or not it is active and to rely on the looping or arrows of interaction to show activity. The PREs should either be left out (not necessary or relevant for this model) or at least made to look different than the two enhancers; otherwise they look like unlabeled enhancers. They were shown clearly enough in an earlier figure and should probably be removed here.

All addressed above.